# Knowledge Distillation Performs Partial Variance Reduction

**Mher Safaryan**
IST Austria
mher.safaryan@ista.ac.at

**Alexandra Peste**
IST Austria
alexandra.peste@ista.ac.at

**Dan Alistarh**
IST Austria
dan.alistarh@ista.ac.at

## Abstract

Knowledge distillation is a popular approach for enhancing the performance of "student" models, with lower representational capacity, by taking advantage of more powerful "teacher" models. Despite its apparent simplicity and widespread use, the underlying mechanics behind knowledge distillation (KD) are still not fully understood. In this work, we shed new light on the inner workings of this method, by examining it from an optimization perspective. We show that, in the context of linear and deep linear models, KD can be interpreted as a novel type of stochastic variance reduction mechanism. We provide a detailed convergence analysis of the resulting dynamics, which hold under standard assumptions for both strongly-convex and non-convex losses, showing that KD acts as a form of *partial variance reduction*, which can reduce the stochastic gradient noise, but may not eliminate it completely, depending on the properties of the "teacher" model. Our analysis puts further emphasis on the need for careful parametrization of KD, in particular w.r.t. the weighting of the distillation loss, and is validated empirically on both linear models and deep neural networks.

## 1 Introduction

Knowledge Distillation (KD) [13, 3] is a standard tool for transferring information between a machine learning model of lower representational capacity–usually called the *student*–and a more accurate and powerful *teacher* model. In the context of classification using neural networks, it is common to consider the student to be a *smaller* network [2], whereas the teacher is a network that is larger and more computationally-heavy, but also more accurate. Assuming a supervised classification task, distillation consists in training the student to minimize the cross-entropy with respect to the *teacher's logits* on every given sample, in addition to minimizing the standard cross-entropy loss with respect to the ground truth labels.

Since its introduction [3], distillation has been developed and applied in a wide variety of settings, from obtaining compact high-accuracy encodings of model ensembles [14], to boosting the accuracy of compressed models [50, 39, 32], to reinforcement learning [51, 43, 36, 5, 7, 46] and learning with privileged information [52]. Given its apparent simplicity, there has been significant interest in finding explanations for the effectiveness of distillation [2, 14, 38]. For instance, one hypothesis [2, 14] is that the smoothed labels resulting from distillation present the student with a decision surface that is *easier to learn* than the one presented by the categorical (one-hot) outputs. Another hypothesis [2, 14, 52] starts from the observation that the teacher's outputs have higher entropy than the ground truth labels, and therefore, higher information content. Despite this work, we still have a limited analytical understanding regarding *why* knowledge distillation is so effective [38]. Specifically, very little is

37th Conference on Neural Information Processing Systems (NeurIPS 2023).

known about the interplay between distillation and stochastic gradient descent (SGD), which is the standard optimization setting in which this method is applied.

**1.1. Contributions.** In this paper, we investigate the impact of knowledge distillation on the convergence of the "student" model when optimizing via SGD. Our approach starts from a simple re-formulation of distillation in the context of gradient-based optimization, which allows us to connect KD to stochastic *variance reduction* techniques, such as SVRG [16], which are popular in stochastic optimization. Our results apply both to *self-distillation*, where KD is applied while training relative to an earlier version of the same model, as well as *distillation for compression*, where a compressed model leverages outputs from an uncompressed one during training.

In a nutshell, in both cases, we show that SGD with distillation preserves the convergence speed of vanilla SGD, but that the teacher's outputs serve to reduce the gradient variance term, proportionally to the distance between the teacher model and the true optimum. Since the teacher model may not be at an optimum, this means that variance reduction is only *partial*, as distillation may not completely eliminate noise, and in fact may introduce bias or even increase variance for certain parameter values. Our analysis precisely characterizes this effect, which can be controlled by the weight of the distillation loss, and is validated empirically for both linear models and deep networks.

**1.2. Results Overview.** To illustrate our results, we consider the case of self-distillation [59], which is a popular supervised training technique in which both the student model $x \in \mathbb{R}^d$ and the teacher model $\theta \in \mathbb{R}^d$ have the same structure and dimensionality. The process starts from a teacher model $\theta$ trained using regular cross-entropy loss, and then trains the student model with respect to a weighted combination of cross-entropy w.r.t. the data labels, and cross-entropy w.r.t. the teacher's outputs. This process is often executed over multiple iterations, in the sense that the student at iteration $k$ becomes the teacher for iteration $k + 1$, and so on.

Our first observation is that, in the case of self-distillation with teacher weight $1 \geq \lambda \geq 0$, the gradient of the distilled student model on a sample $i$, denoted by $\nabla_x f_i(x \mid \theta, \lambda)$ at a given iteration can simply be written as

$$\nabla_x f_i(x \mid \theta, \lambda) \simeq \nabla f_i(x) - \lambda \nabla f_i(\theta),$$

where $\nabla f_i(x)$ and $\nabla f_i(\theta)$ are the student's and teacher's standard gradients on sample $i$, respectively. This expression is exact for linear models and generalized linear networks, and we provide evidence that it holds approximately for general classifiers. With this re-formulation, self-distillation can be interpreted as a truncated form of the classic SVRG iteration [16], which never employs full (non-stochastic) gradients.

Our main technical contribution is in providing convergence guarantees for iterations of this form, for both strong quasi-convex, and non-convex functions under the Polyak-Łojasiewicz (PL) condition. Our analysis covers both self-distillation (where the teacher is a partially-trained version of the same model), and more general distillation, in which the student is a compressed version of the teacher model. The convergence rates we provide are similar in nature to SGD convergence, with one key difference: the rate dependence on the *gradient variance* $\sigma^2$ is dampened by a term depending on the gap between the teacher model and an optimum. Intuitively, this says that, if the teacher's accuracy is poor, then distillation will not have any positive effect. However, the better-trained the teacher is, the more it can help reduce the *student's* variance during optimization. Importantly, this effect occurs even if the teacher is *not* trained to near-zero loss, thus motivating the usefulness of the teacher in self-distillation. Our analysis highlights the importance of the *distillation weight*, as a means to maximize the positive effects of distillation: for linear models, we can even derive a closed-form solution for the optimal distillation weight. We validate our findings experimentally for both linear models and deep networks, confirming the effects predicted by the analysis.

## 2  Related Work

We now provide an overview for some of the relevant related work regarding KD. Knowledge distillation, in its current formulation, was introduced in the seminal work of [13], which showed that the predictive performance of a model can be improved if it is trained to match the soft targets produced by a large and accurate model. This observation has motivated the adoption of KD as a standard mechanism to enhance the training of neural networks in a wide range of settings, such as compression [39], learning with noisy labels [26], and has also become an essential tool in training accurate compressed versions of large models [39, 49, 54].

Despite these important practical advantages, providing a thorough theoretical justification for the mechanisms driving the success of KD has so far been elusive. Several works have focused on studying KD from different theoretical perspectives. For example, Lopez et al. [28] connected distillation with *privileged information* [52] by proposing the notion of *generalized distillation*, and presented an intuitive explanation for why generalized distillation should allow for higher sample efficiency, relative to regular training. Phuong and Lampert [38] studied distillation from the perspective of generalization bounds in the case of linear and deep linear models. They identify three factors which influence the success of KD: (1) the geometry of the data; (2) the fact that the expected risk of the student always decreases with more data; (3) the fact that gradient descent finds a favorable minimum of the distillation objective. By contrast to all these previous references, our work studies the impact of distillation on stochastic (SGD-based) optimization.

More broadly, there has been extensive work on connecting KD with other areas in learning. Dao et al. [6] examined links between KD and semi-parametric Inference, whereas Li et al. [61] performed an empirical study on KD in the context of learning with noisy labels. Yuan et al. [58] and Sultan et al. [48] investigated the relationships between KD and label smoothing, a popular heuristic for training neural networks, showing both similarities and substantive differences. In this context, our work is the first to signal the connection between KD and variance-reduction, as well as investigating the convergence of KD in the context of stochastic optimization.

## 3  Knowledge Distillation

**3.1. Background.** Assume we are given a finite dataset $\{(a_n, b_n) \mid n = 1, 2, \ldots, N\}$, where inputs $a_n \in \mathcal{A}$ (e.g., vectors from $\mathbb{R}^d$) and outputs $b_n \in \mathcal{B}$ (e.g., categorical labels or real numbers). Consider a set of models $\mathcal{F} = \{\phi_x : \mathcal{A} \to \mathcal{B} \mid x \in \mathcal{P} \subseteq \mathbb{R}^d\}$ with fixed neural network architecture parameterized by vector $x$. Depending on the supervised learning task and the class $\mathcal{F}$ of models, we define a loss function $\ell \colon \mathcal{B} \times \mathcal{B} \to \mathbb{R}_+$ in order to measure the performance of the model. In particular, the loss associated with a data point $(a_n, b_n)$ and model $\phi_x \in \mathcal{F}$ would be $\ell(\phi_x(a_n), b_n)$. In this framework, the standard Empirical Risk Minimization (ERM) takes the following form:

$$\min_{x \in \mathbb{R}^d} \frac{1}{N} \sum_{n=1}^{N} \ell(\phi_x(a_n), b_n). \tag{1}$$

In the objective above, the model $\phi_x$ is trained to match the true outputs $b_n$ given in the training dataset. Suppose that in addition to the true labels $b_n$, we have access to sufficiently well-trained and perhaps more complicated teacher model's outputs $\Phi_\theta(a_n) \in \mathcal{B}$ for each input $a_n \in \mathcal{A}$. Similar to the student model $\phi_x$, the teacher model $\Phi_\theta$ maps $\mathcal{A} \to \mathcal{B}$ but can have different architecture, more layers and parameters. The fundamental question is how to exploit the additional knowledge of the teacher $\Phi_\theta$ to facilitate the training of a more compact student model $\phi_x$ with lower representational capacity. *Knowledge Distillation* with parameter $\lambda \in [0, 1]$ from teacher model $\Phi_\theta$ to student model $\phi_x$ is the following modification to the objective (1):

$$\min_{x \in \mathbb{R}^d} \frac{1}{N} \sum_{n=1}^{N} \Big[ (1 - \lambda) \ell(\phi_x(a_n), b_n) + \lambda \ell(\phi_x(a_n), \Phi_\theta(a_n)) \Big]. \tag{2}$$

Here we customize the loss penalizing dissimilarities from the teacher's feedback $\Phi_\theta(a_n)$ in addition to the true outputs $b_n$. In case of $\ell$ is linear in the second argument (e.g., cross-entropy loss), the problem simplifies into

$$\min_{x \in \mathbb{R}^d} \frac{1}{N} \sum_{n=1}^{N} \ell(\phi_x(a_n), (1 - \lambda) b_n + \lambda \Phi_\theta(a_n)), \tag{3}$$

which is a standard ERM (1) with modified "soft" labels $s_n := (1 - \lambda) b_n + \lambda \Phi_\theta(a_n)$ as the target.

**3.2. Self-distillation.** As already mentioned, the teacher's model $\Phi_\theta$ can have more complicated neural network architecture and potentially larger parameter space $\theta \in \mathcal{Q} \subseteq \mathbb{R}^D$. In particular, $\Phi_\theta$ does not have to be from the same set of models $\mathcal{F}$ as the student model $\phi_x$. The special case when both the student and the teacher share the same structure/architecture is called *self-distillation* [33, 60], which is the key setup for our work. In this case, the teacher model $\Phi_\theta \equiv \phi_\theta \in \mathcal{F}$ with $\theta \in \mathbb{R}^d$ (i.e., $\mathcal{Q} = \mathcal{P}$, $D = d$) and the corresponding distillation objective would be

$$\min_{x \in \mathbb{R}^d} \frac{1}{N} \sum_{n=1}^{N} \Big[ (1 - \lambda) \ell(\phi_x(a_n), b_n) + \lambda \ell(\phi_x(a_n), \phi_\theta(a_n)) \Big]. \tag{4}$$

---
**Algorithm 1** Knowledge Distillation via SGD
---
1: **Input:** learning rate $\gamma > 0$, initial student model $x^0 \in \mathcal{P} \subseteq \mathbb{R}^d$
2: **for** each distillation iteration $m$ **do**
3:     choose a teacher model $\theta^m \in \mathcal{Q} \subseteq \mathbb{R}^D$ and distillation weight $\lambda^m \in [0, 1]$ (e.g., see Sec. 5)
4:     **for** each training iteration $t$ **do**
5:         sample an unbiased mini-batch $\xi \sim \mathcal{D}$ form the train set
6:         compute distillation gradient $\nabla f_\xi(x^t \mid \theta^m, \lambda^m) = \lambda^m \nabla f_\xi(x^t) + (1 - \lambda^m) \nabla f_\xi(x^t \mid \theta^m)$
7:         update the student model via $x^{t+1} = x^t - \gamma \nabla f_\xi(x^t \mid \theta^m, \lambda^m)$
8:     **end for**
9: **end for**
---

Our primary focus in this work would be the objective mentioned above of self-distillation. For convenience, let $f_n(x) := \ell(\phi_x(a_n), b_n)$ be the prediction loss with respect to the output $b_n$, $f_n(x \mid \theta) := \ell(\phi_x(a_n), \phi_\theta(a_n))$ be the loss with respect to the teacher's output probabilities and $f_n(x \mid \theta, \lambda) := \lambda f_n(x) + (1 - \lambda) f_n(x \mid \theta)$ be the distillation loss. See Algorithm 1 for an illustration.

**3.3. Distillation Gradient.** As the first step towards understanding how self-distillation affects the training procedure, we analyze the modified loss landscape (4) via stochastic gradients of (1) and (4). In particular, we put forward the following proposition regarding the form of distillation gradient in terms of gradients of (1).

**Proposition 1** (Distillation Gradient)**.** *For a student model $x \in \mathbb{R}^d$, teacher model $\theta \in \mathbb{R}^d$ and distillation weight $\lambda$, the distillation gradient corresponding to self-distillation* (4) *is given by*

$$\nabla_x f_n(x \mid \theta, \lambda) = \nabla f_n(x) - \lambda \nabla f_n(\theta). \tag{5}$$

Before justifying this proposition formally, let us provide some intuition behind the expression (5) and its connection to distillation loss (4). First, the gradient expression (5) suggests that the teacher has little or no effect on the data points classified correctly with high confidence (i.e. those for which $\nabla f_n(\theta)$ is close to 0 or $\phi_\theta(a_n)$ is close to $b_n$). In other words, the more accurate the teacher is, the less it can affect the learning process. In the extreme case, a perfect or overfitted teacher (one that $\nabla f_n(\theta) = 0$ or $\phi_\theta(a_n) = b_n$ for all $n$) will have no effect. In fact, this is expected since, in this case, problems (1) and (4) coincide. Alternatively, if the teacher is not perfect, then the modified objective (4) intuitively suggests that the learning dynamics of a student model is adjusted based on the teacher's knowledge. As we can see in (5), the adjustment from the teacher is enforced by $-\lambda \nabla f_n(\theta)$ term. It is worth mentioning that the direction of distillation gradient $\nabla_x f_n(x \mid \theta, \lambda)$ can be different from the usual gradient's direction $\nabla f_n(x)$ due to the influence of the teacher. Thus, Proposition 1 explicitly shows how the teacher guides the student by adjusting its stochastic gradient.

As we will show later, distillation gradient (5) leads to partial variance reduction because of the additional $-\lambda \nabla f_\xi(\theta)$ term. When chosen properly (distillation weight $\lambda$ and proximity of $\theta$ to the optimal solution $x^*$), this additional stochastic gradient is capable of adjusting the student's stochastic gradient since both are computed using the same batch from the train data. In other words, both gradients have the same source of randomness which makes partial cancellations feasible.

• **Linear regression.** To support Proposition 1 rigorously, consider the simple setup of linear regression. Let $\mathcal{A} = \mathbb{R}^d$, $\mathcal{P} = \mathbb{R}^{d+1}$, $\phi_x(a) = x^\top \bar{a} \in \mathbb{R}$, where $\bar{a} = [a \; 1]^\top \in \mathbb{R}^{d+1}$ is the input vector in the lifted space (to include the bias term), $\mathcal{B} = \mathbb{R}$, and the loss is defined by $\ell(t, t') = (t - t')^2$ for all $t, t' \in \mathbb{R}$. Thus, based on (4), we have

$$f_n(x \mid \theta, \lambda) \;\; = \;\; (1 - \lambda)(x^\top \bar{a}_n - b_n)^2 + \lambda(x^\top \bar{a}_n - \theta^\top \bar{a}_n)^2,$$

from which we compute its gradient straightforwardly as

$$\begin{aligned} \nabla_x f_n(x \mid \theta, \lambda) \;\; &= \;\; 2(1 - \lambda)(x^\top \bar{a}_n - b_n)\bar{a}_n + 2\lambda(x^\top \bar{a}_n - \theta^\top \bar{a}_n)\bar{a}_n \\ &= \;\; 2(x^\top \bar{a}_n - b_n)\bar{a}_n - 2\lambda(\theta^\top \bar{a}_n - b_n)\bar{a}_n = \nabla f_n(x) - \lambda \nabla f_n(\theta). \end{aligned}$$

Hence, the distillation gradient for linear regression tasks has the form (5).

• **Classification with a single hidden layer.** We can extend the above argument and derivation for a $K$-class classification model with one hidden layer that has soft-max as the last layer, i.e. $\phi_X(a_n) = \sigma(X^\top a_n) \in \mathbb{R}^K$, where $X = [x_1 \; x_2 \; \ldots \; x_K] \in \mathbb{R}^{d \times K}$ are the model parameters,

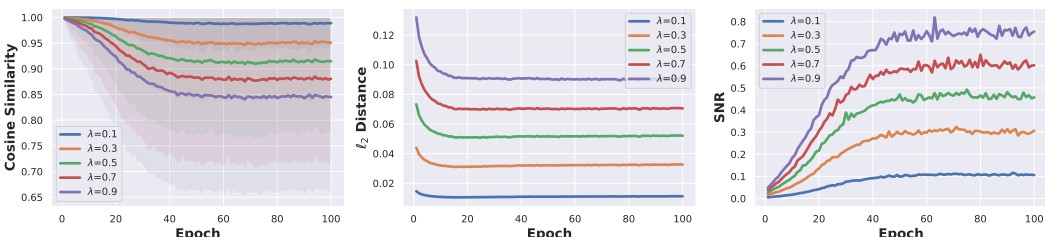

Figure 1: Cosine similarity, $l_2$ distance and SNR (i.e., $l_2$ distance over the gradient norm of standard KD) statistics between the true and approximated distillation gradient for a neural network during training. As predicted, larger $\lambda$ leads to larger differences, although gradients remain well-correlated.

$a_n \in \mathcal{A} = \mathbb{R}^d$ is the input data and $\sigma$ is the soft-max function. Then, we show in the Appendix B.2 that for all $k = 1, 2 \ldots, K$ it holds

$$\nabla_{x_k} f_n(X \mid \Theta, \lambda) = \nabla_{x_k} f_n(X) - \lambda \nabla_{\theta_k} f_n(\Theta),$$

where $\Theta = [\theta_1 \; \theta_2 \; \ldots \; \theta_K] \in \mathbb{R}^{d \times K}$ are the teacher's parameters.

• **Generic classification.** Proposition 1 will not hold precisely for arbitrary deep non-linear neural networks. However, careful calculations reveal that, in general, distillation gradient takes a form similar to (5). Detailed derivations are deferred to Appendix B.3, here we provide the sketch.

Consider an arbitrary neural network architecture for classification that ends with soft-max layer, i.e. $\phi_x(a_n) = \sigma(\psi_n(x))$, where $a_n \in \mathcal{A}$ is the input data, $\psi_n(x)$ are the produced logits with respect to the model parameters $x$, and $\sigma$ is the soft-max function. Denote $\varphi_n(z) := \ell(\sigma(z), b_n)$ the loss associated with logits $z$ and the true label $b_n$. In words, $\psi_n$ gives the logits from the input data, while $\varphi_n$ computes the loss from given logits. Then, the representation for the loss function is $f_n(x) = \varphi_n(\psi_n(x))$. We show in Appendix B.3 that the distillation gradient can be written as

$$\nabla_x f_n(x \mid \theta, \lambda) = J\psi_n(x)\left(\nabla\varphi_n(\psi_n(x)) - \lambda\nabla\varphi_n(\psi_n(\theta))\right) = \frac{\partial\psi_n(x)}{\partial x}\frac{\partial f_n(x)}{\partial\psi_n(x)} - \lambda\frac{\partial\psi_n(x)}{\partial x}\frac{\partial f_n(\theta)}{\partial\psi_n(\theta)},$$

where $J\psi_n(x) := \frac{\partial\psi_n(x)}{\partial x} = [\nabla\psi_{n,1}(x) \; \nabla\psi_{n,2}(x) \ldots \nabla\psi_{n,K}(x)] \in \mathbb{R}^{d \times K}$ is the Jacobian of the vector-valued function $\psi_n$ for logits. Notice that the first term $\frac{\partial\psi_n(x)}{\partial x}\frac{\partial f_n(x)}{\partial\psi_n(x)}$ coincides with the student's gradient $\nabla_x f_n(x) = \frac{\partial f_n(x)}{\partial x}$. However, the second term $\frac{\partial\psi_n(x)}{\partial x}\frac{\partial f_n(\theta)}{\partial\psi_n(\theta)}$ differs from the teacher's gradient as the partial derivatives of logits are with respect to the student model.

Despite these differences in the case of deep non-linear models, we observe that the distillation gradient defined by Equation 5 can approximate well the true distillation gradient from Equation 3. Specifically, we consider a fully connected neural network with one hidden layer and ReLU activation [34], trained on the MNIST dataset [24], using regular self-distillation, from an SGD-teacher, with a fixed learning rate, and SGD with weight decay and no momentum. At each training iteration we compute the cosine similarity between the gradient of the distillation loss and the approximation from Equation 5, and we average the results across each epoch. The results presented in Figure 1 show that the distillation gradient approximates well the true distillation gradient. Moreover, the behavior is monotonic in the distillation weight $\lambda$ (higher similarity for smaller $\lambda$), as predicted by the analysis above, and it stabilizes as training progresses. The decrease of cosine similarity can be explained as follows: at the beginning the cosine similarity is high (and SNR is low) since we start from the same model. Then, initial perturbations caused by either the KD or modified KD gradient don't cause big shifts (the teacher has enough confidence and small gradients). These perturbations accumulate over the training leading to decreased cosine similarity and eventually stabilize.

## 4 Convergence Theory for Self-Distillation

**4.1. Optimization Setup and Assumptions.** We abstract the standard ERM problem (1) into a stochastic optimization problem of the form

$$\min_{x \in \mathbb{R}^d}\left\{f(x) := \mathbb{E}_{\xi \sim \mathcal{D}}\left[f_\xi(x)\right]\right\}, \tag{6}$$

where $f_\xi(x)$ is the loss associated with data sample $\xi \sim \mathcal{D}$ given model parameters $x \in \mathbb{R}^d$. For instance, if $\xi = (a_n, b_n)$ is a single data point, then the corresponding loss is $f_\xi(x) = \ell(\phi_x(a_n), b_n)$. The goal is to find parameters $x$ minimizing the risk $f(x)$. To solve the problem (6), we employ *Stochastic Gradient Descent (SGD)*. Based on Section 3, applying SGD to the problem (6) with self-distillation amounts to the following optimization updates in the parameter space:

$$x^{t+1} = x^t - \gamma(\nabla f_\xi(x^t) - \lambda \nabla f_\xi(\theta)), \tag{7}$$

with initialization $x^0 \in \mathbb{R}^d$, step size or learning rate $\gamma > 0$, teacher model's parameters $\theta \in \mathbb{R}^d$ and distillation weight $\lambda$. To analyze the convergence behavior of iterates (7), we need to impose some assumptions in order to derive reasonable convergence guarantees. First, we assume that the problem (6) has a non-empty solution set $\mathcal{X} \neq \emptyset$ and $f^* := f(x^*)$ for some minimizer $x^* \in \mathcal{X}$.

**Assumption 1** (Strong quasi-convexity). *The function $f : \mathbb{R}^d \to \mathbb{R}$ is differentiable and $\mu$-strongly quasi-convex for some constant $\mu > 0$, i.e., for any $x \in \mathbb{R}^d$ it holds*

$$f(x^*) \geq f(x) + \langle \nabla f(x), x^* - x \rangle + \frac{\mu}{2}\|x^* - x\|^2. \tag{8}$$

Strong quasi-convexity [9] is a weaker version of strong convexity [35], which assumes that the quadratic lower bound above holds for at every point $y \in \mathbb{R}^d$ instead of $x^* \in \mathcal{X}$. Notice that strong quasi-convexity implies that the minimizer $x^*$ is unique[1]. A more relaxed version of this assumption is the Polyak-Łojasiewicz (PL) condition [40].

**Assumption 2** (Polyak-Łojasiewicz condition). *Function $f : \mathbb{R}^d \to \mathbb{R}$ is differentiable and satisfies PL condition with parameter $\mu > 0$, if for any $x \in \mathbb{R}^d$ it holds*

$$\|\nabla f(x)\|^2 \geq 2\mu(f(x) - f^*). \tag{9}$$

Note that the requirement imposed by the PL condition above is weaker than by strong convexity. Functions satisfying PL condition do not have to be convex and can have multiple minimizers [17]. We make use of the following form of smoothness assumption on the stochastic gradient commonly referred to as *expected smoothness* in the optimization literature [11, 10, 19].

**Assumption 3** (Expected Smoothness). *Functions $f_\xi(x)$ are differentiable and $\mathcal{L}$-smooth in expectation with respect to subsampling $\xi \sim \mathcal{D}$, i.e., for any $x \in \mathbb{R}^d$ it holds*

$$\mathbb{E}_{\xi \sim \mathcal{D}}\left[\|\nabla f_\xi(x) - \nabla f_\xi(x^*)\|^2\right] \leq 2\mathcal{L}(f(x) - f^*) \tag{10}$$

*for some constant $\mathcal{L} = \mathcal{L}(f, \mathcal{D})$.*

The expected smoothness condition above is a joint property of loss function $f$ and data subsampling strategy from the distribution $\mathcal{D}$. In particular, it subsumes the smoothness condition for $f(x)$ since (10) also implies $\|\nabla f(x) - \nabla f(x^*)\|^2 \leq 2\mathcal{L}(f(x) - f^*)$ for any $x \in \mathbb{R}^d$. We denote by $L$ the smoothness constant of $f(x)$ and notice that $L \leq \mathcal{L}$.

**4.2. Convergence Theory and Partial Variance Reduction.** Equipped with the assumptions described in the previous part, we now present our convergence guarantees for the iterates (7) for both strong quasi-convex and PL loss functions.

**Theorem 1** (See Appendix C.2). *Let Assumptions 1 and 3 hold. For any $\gamma \leq \frac{1}{8\mathcal{L}}$ and properly chosen distillation weight $\lambda$, the iterates (7) of SGD with self-distillation using teacher's parameters $\theta$ converge as*

$$\mathbb{E}\left[\|x^t - x^*\|^2\right] \leq (1 - \gamma\mu)^t \|x^0 - x^*\|^2 + \frac{2\sigma_*^2}{\mu}\min(\gamma, \mathcal{O}(f(\theta) - f^*)), \tag{11}$$

*where $\sigma_*^2 := \mathbb{E}[\|\nabla f_\xi(x^*)\|^2]$ is the stochastic noise at the optimum.*

**Theorem 2** (See Appendix C.3). *Let Assumptions 2 and 3 hold. For any $\gamma \leq \frac{1}{4\mathcal{L}}\frac{\mu}{L}$ and properly chosen distillation weight $\lambda$, the iterates (7) of SGD with self-distillation using teacher's parameters $\theta$ converge as*

$$\mathbb{E}\left[f(x^t) - f^*\right] \leq (1 - \gamma\mu)^t \left(f(x^0) - f^*\right) + \frac{L\sigma_*^2}{\mu}\min(\gamma, \mathcal{O}(f(\theta) - f^*)), \tag{12}$$

---

[1]If $f(x^*) = f(x^{**})$, then $\frac{\mu}{2}\|x^* - x^{**}\| \leq f(x^*) - f(x^{**}) = 0$. Hence, $x^* = x^{**}$.

*Proof overview.* Both proofs follow similar steps and can be divided into three logical parts.

*Part 1 (Descent inequality).* Generally, an integral part of essentially any convergence theory for optimization methods is a descent inequality quantifying the progress of an algorithm in one iteration. Our theory is not an exception: we first define our "potential" $e^t = \|x^t - x^*\|^2$ for the strongly quasi-convex setup, and $e^t = f(x^t) - f^*$ for the PL setup. Then, we start our derivations by bounding $\mathbb{E}_t[e^{t+1}] - (1 - \gamma\mu)e^t$. Here, $\mathbb{E}_t$ is the conditional expectation with respect the randomness of previous iterate $x^t$. Specifically, up to constants, both setups allow the following bound:

$$\mathbb{E}_t[e^{t+1}] \leq (1 - \gamma\mu)e^t - \mathcal{O}(\gamma)(1 - \mathcal{O}(\gamma))(f(x^t) - f^*) + \mathcal{O}(\gamma)N(\lambda), \tag{13}$$

where $N(\lambda) = \lambda^2\|\nabla f(\theta)\|^2 + \gamma\mathbb{E}\left[\|\nabla f_\xi(x^*) - \lambda\nabla f_\xi(\theta)\|^2\right]$. Choosing the learning rate $\gamma$ to be small enough, we ensure that the second term is non-positive and hence negligible in the upper bound.

*Part 2 (Optimal distillation weight).* Next, we focus our attention on the third term in (13) involving the iteration-independent neighborhood term $N(\lambda)$. Note that the $\mathcal{O}(\gamma)$ factor next to $N(\lambda)$ will be absorbed once we unfold the recursion (13) up to initial iterate. Hence, the convergence neighborhood is proportional to $N(\lambda)$. Now the question is how small this term can get if we properly tune the parameter $\lambda$. Notice that $N(0) = \gamma\sigma_*^2$ corresponds to the neighborhood size for plain SGD without any distillation involved. Luckily, due to the quadratic dependence, we can minimize $N(\lambda)$ analytically with respect to $\lambda$ and find the optimal value

$$\lambda_* = \frac{\mathbb{E}[\langle\nabla f_\xi(x^*), \nabla f_\xi(\theta)\rangle]}{\mathbb{E}[\|\nabla f_\xi(\theta)\|^2] + \frac{1}{\gamma}\|\nabla f(\theta)\|^2}. \tag{14}$$

Consequently, the analysis puts further emphasis on the need for careful parametrization with respect to the weighting $\lambda$ of the distillation loss as there exists a particularly privileged value $\lambda_*$.

*Part 3 (Impact of the teacher).* In the final step, we quantify the impact of the teacher on the reduction in the neighborhood term $N(\lambda_*)$ compared to the plain SGD neighborhood $N(0)$. Via algebraic transformations, we show that

$$\frac{N(\lambda_*)}{N(0)} = 1 - \rho^2(x^*, \theta)\frac{1 - \beta(\theta)}{1 + \frac{1}{\gamma}\beta(\theta)}, \tag{15}$$

where $\beta(\theta) = \|\nabla f(\theta)\|^2 / \mathbb{E}[\|\nabla f_\xi(\theta)\|^2] \in [0, 1]$ is the signal-to-noise ratio, and $\rho(x^*, \theta) \in [-1, 1]$ is the correlation coefficient between stochastic gradients $\nabla f_\xi(x^*)$ and $\nabla f_\xi(\theta)$. This representation gives us analytical means to measure the impact of the teacher. For instance, the optimal teacher $\theta = x^*$ satisfies $\rho(x^*, \theta) = 1$ and $\beta(\theta) = 0$, and thus $N(\lambda_*) = 0$. In general, if the teacher is not optimal, then the reduction (15) is of order $\frac{1}{\gamma}\mathcal{O}(f(\theta) - f^*)$ and $N(\lambda_*) = \sigma_*^2 \min(\gamma, \mathcal{O}(f(\theta) - f^*))$. $\qquad\square$

We discuss these results highlighting the key aspects and significance.

● **Structure of the rates.** The structure of these two convergence rates is typical in gradient-based stochastic optimization literature: linear convergence up to some neighborhood controlled by the stochastic noise term $\sigma_*^2$ and learning rate $\gamma$. In fact, these results (including learning rate restrictions) are identical to ones for SGD [8] except the non-vanishing terms in (11) and (12) include an additional $\mathcal{O}(f(\theta) - f^*)$ factor due to distillation and proper selection of weight parameter $\lambda$. For both setups, the rate of SGD is the same (11) or (12) with only one difference: $\min(\gamma, \mathcal{O}(f(\theta) - f^*))$ term is replaced with $\gamma$. So, $\mathcal{O}(f(\theta) - f^*)$ is the factor that makes our results better compared to SGD in terms of optimization performance.

● **Importance of the results.** First, observe that in the worst case when the teacher's parameters are trained inadequately (or not trained at all), that is $\mathcal{O}(f(\theta) - f^*) \geq \gamma$, then the obtained rates recover the known results for plain SGD. However, the crucial benefit of these results is to show that a sufficiently well-trained teacher, i.e. $\mathcal{O}(f(\theta) - f^*) < \gamma$, provably reduces the neighborhood size of SGD without slowing down the speed of convergence. In the best case scenario, when the teacher's model is perfectly trained, namely $f(\theta) = f^*$, then the neighborhood term vanishes, and the method converges to the exact solution (see SGD-star Algorithm 4 in [9]). Thus, self-distillation in the form of iterates (7) acts as a form of *partial variance reduction*, which can reduce the stochastic gradient noise, but may not eliminate it completely, depending on the properties of the teacher model.

● **Choice of distillation weight $\lambda$.** As we discussed in the proof sketch above, our analysis reveals that the performance of distillation is optimized for a specific value (14) of distillation weight $\lambda_*$ depending on the teacher model. One way to interpret the expression (14) for weight parameter

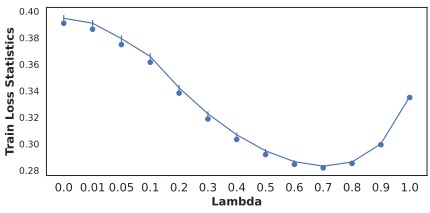
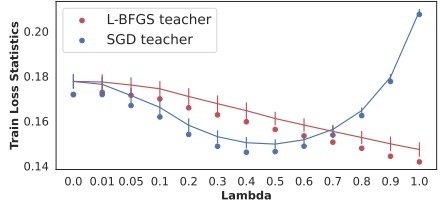

(a) Self-distillation on MNIST.

(b) Self-distillation on CIFAR-10.

Figure 2: The minimum training loss, and average over the last 10 epochs, for models trained with SGD and with self-distillation, using different values of the distillation parameter $\lambda$, on MNIST and CIFAR-10. SGD is equivalent to $\lambda = 0$. The curves for the SGD-based teachers (which do not have zero loss) reflect our analysis, corroborating the existence of the "optimal" distillation weight. By contrast, in the L-BFGS teacher, higher distillation weight always leads to lower loss.

intuitively is that the better the teacher's model $\theta$ is, the bigger $\lambda_* \leq 1$ gets. In other words, $\lambda_*$ quantifies the quality of the teacher: $\lambda_* \approx 0$ indicates a poor teacher model ($f(\theta) \gg f^*$) and $\lambda_* = 1$ is for the optimal teacher ($f(\theta) = f^*$).

**4.3. Experimental Validation.** In this section we illustrate that our theoretical analysis in the convex case also holds empirically. Specifically, we consider classification problems using linear models in two different setups: training a linear model on the MNIST dataset [24] and linear probing on the CIFAR-10 dataset [23], using a ResNet50 model [12], pre-trained on the ImageNet dataset [42]. For the second setup we train a linear classifier on top of the features extracted from a ResNet50 model pre-trained on ImageNet. This is a standard setting, commonly used in the transfer learning literature, see e.g. [21, 45, 15]. In both cases we train using SGD without momentum and regularization, with a fixed learning rate and mini-batch of size 10, for a total of 100 epochs. The models trained with SGD are compared against self-distillation (Equation 7), using the same training hyper-parameters, where the teacher is the model trained with SGD. In the case of CIFAR-10 features, we also consider the "optimal" teacher, which is a model trained with L-BFGS [27]. We perform all experiments using multiple values of the distillation parameter $\lambda$ and measure the cross entropy loss between student and true labels. At each training epoch we computing the running average over all mini-batch losses seen during that epoch.

The results presented in Figure 2 show the minimum cross entropy train loss obtained over 100 epochs, as well as the average over the last 10 epochs, for models trained with SGD, as well as with self-distillation, with $\lambda \in [0.01, 1]$. We observe that when the teacher is the model trained with SGD ($\lambda = 0$), there exists a $\lambda > 0$ which achieves a lower training loss than SGD, which is in line with our statement from Theorem 1. Furthermore, when the teacher is very close to the optimum, $\lambda$ closer to 1 reduces the training loss the most compared to SGD, which is also in line with the theory (see Theorem 1). This behavior is illustrated in Figure 2b, when using an L-BFGS teacher.

## 5 Removing Bias and Improving Variance Reduction

In this section, we investigate the cause of having variance reduction only partially and suggest a possible workaround to obtain complete variance reduction. In brief, the potential source of *partial* variance reduction is the biased nature of distillation. Essentially, distillation bias is reflected in the iterates (7) since the expected update direction $\mathbb{E}\left[\nabla f_\xi(x^t) - \lambda \nabla f_\xi(\theta) \mid x^t\right] = \nabla f(x^t) - \lambda \nabla f(\theta)$ can be different from $\nabla f(x^t)$. This comes from the fact that distillation loss (4) modifies the initial loss (1) composed of true outputs. To make our argument compelling, next we correct the bias by adding $\lambda \nabla f(\theta)$ to iterates (7) and analyze the following dynamics:

$$x^{t+1} = x^t - \gamma(\nabla f_\xi(x^t) - \lambda \nabla f_\xi(\theta) + \lambda \nabla f(\theta)). \tag{16}$$

Besides making the estimate unbiased, the advantage of this adjustment is that no tuning is required for the distillation weight $\lambda$; we may simply set $\lambda = 1$. The obvious disadvantage is that $\nabla f(\theta)$ is the batch gradient over the whole train data that can be very costly to compute. However, we could compute it once and reuse it for all further iterates. The resulting iteration is similar to the popular and well-studied SVRG [16, 57, 41, 20, 25, 22, 29] method, and therefore iterates (16) will enjoy full variance reduction.

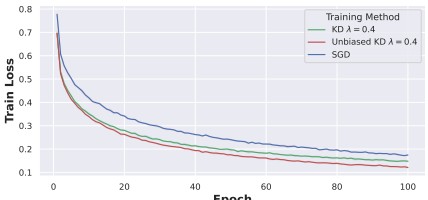 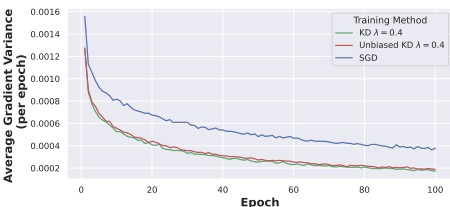

Figure 3: (Left plot) The train loss of self-distillation, unbiased self-distillation and vanilla SGD training. (Right plot) The progress of gradient variances (averaged over the iterations within each epoch) for the same setup

**Theorem 3** (See Appendix D). *Let Assumptions 2 and 3 hold. Then for any $\gamma \leq \frac{\mu}{3L\mathcal{L}}$ the iterates* (16) *with $\lambda = 1$ converge as*

$$\mathbb{E}_t\left[f(x^t) - f^*\right] \leq (1 - \gamma\mu)^t (f(x^0) - f^*) + \frac{3L(L+\mathcal{L})}{\mu} \cdot \gamma \left(f(\theta) - f^*\right). \tag{17}$$

The key improvement that bias correction brings in (17) is the convergence up to a neighborhood $\mathcal{O}(\gamma(f(\theta) - f^*))$ in contrast to $\min(\gamma, \mathcal{O}(f(\theta) - f^*))$ as in (11) and (12). The multiplicative dependence of learning rate and the quality of the teacher leads the method (16) to full variance reduction. Indeed, if we choose the teacher model as $\theta = x^0$, then the rate (17) becomes

$$\mathbb{E}\left[f(x^t) - f^*\right] \leq \left[(1 - \gamma\mu)^t + \gamma \cdot 3L(L + \mathcal{L})/\mu\right] (f(x^0) - f^*) \leq \tfrac{1}{2}(f(x^0) - f^*),$$

provided sufficiently small step-size $\gamma \leq \frac{\mu}{12L\mathcal{L}}$ and enough training iterations $t = \mathcal{O}(1/\gamma\mu)$. Hence, following SVRG and updating the teacher model in every $\tau = \mathcal{O}(1/\gamma\mu)$ training iterations, that is choosing $\theta^m = x^{m\tau}$ as the teacher at the $m^{th}$ distillation iteration (see line 3 of Algorithm 1), we have

$$\mathbb{E}\left[f(x^{m\tau}) - f^*\right] \leq \tfrac{1}{2}(f(x^{(m-1)\tau}) - f^*) \leq \tfrac{1}{2^m}(f(x^0) - f^*).$$

Thus, we need $\mathcal{O}(\log \frac{1}{\epsilon})$ bias-corrected distillation iteration phases, each with $\tau = \mathcal{O}(1/\gamma\mu)$ training iterates, to get $\epsilon$ accuracy in function value. Overall, this amounts to $\mathcal{O}(\frac{1}{\gamma\mu} \log \frac{1}{\epsilon})$ iterations of (16).

**Experimental Validation.** Similarly to the previous section, we further validate empirically the result from Theorem 3. Specifically, we consider the convex setup described before, where we train linear models on features extracted on the CIFAR-10 dataset. Based on Figure 2b, we select $\lambda = 0.4$ achieving the largest reduction in train loss, compared to SGD, and we additionally perform unbiased self-distillation (Equation 16), using the same training hyperparameters. Similar to the setup from Figure 2, we measure the cross entropy train loss of the student and with the true labels, which is computed at each epoch by averaging the mini-batch losses. The results are averaged over three runs and presented in Figure 3. The first plot on the left shows that, indeed, the unbiased self-distillation update further reduces the training loss, compared to the update from Equation 7. The second plot explicitly tracks gradient variance (averaged over the iterations within each epoch) for the same setup. As expected, both variants of KD (biased and unbiased) have reduced gradient variance compared to plain SGD. The plot also highlights that both variants of KD have similar variance reduction properties, while the unbiasedness of unbiased KD amplifies the reduction of train loss.

## 6 Convergence for Distillation of Compressed Models

So far, the theory we have presented is for self-distillation, i.e., the teacher's and student's architectures are identical. To understand the impact of knowledge distillation, we relax this requirement and allow the student's model to be a sub-network of the larger and more powerful teacher's model. Our approach to model this relationship between the student and the teacher is to view the student as a masked or, in general, compressed version of the teacher. Hence, as an extension to (7) we analyze the following dynamics of distillation with compressed iterates:

$$x^{t+1} = \mathcal{C}(x^t - \gamma(\nabla f_\xi(x^t) - \lambda\nabla f_\xi(\theta))), \tag{18}$$

where student's parameters are additionally compressed in each iteration using an unbiased compression operator defined below.

**Assumption 4.** *The compression operator* $\mathcal{C} : \mathbb{R}^d \to \mathbb{R}^d$ *is unbiased and there exists finite* $\omega \geq 0$ *bounding the compression variance variance, i.e., for all* $x \in \mathbb{R}^d$ *we have*

$$\mathbb{E}[\mathcal{C}(x)] = x, \qquad \mathbb{E}[\|\mathcal{C}(x) - x\|^2] \leq \omega \|x\|^2. \qquad (19)$$

Typical examples of compression operators satisfying conditions (19) are sparsification [55, 47] and quantization [1, 56], which are heavily used in the context of communication efficient distributed optimization and federated learning [30, 44, 37, 53]. In this context, we obtain the following:

**Theorem 4** (See Appendix E). *Let smoothness Assumption 3 hold and* $f$ *be* $\mu$-*strongly convex. Choose any* $\gamma \leq \frac{1}{16\mathcal{L}}$ *and compression operator with variance parameter* $\omega = \mathcal{O}(\mu/\mathcal{L})$. *Then, properly selecting distillation weight* $\lambda$, *the iterates* (18) *satisfy*

$$\mathbb{E}\left[\|x^t - x^*\|^2\right] \leq \mathcal{O}(\omega + 1)\left[(1 - \gamma\mu)^t\|x^0 - x^*\|^2 + \frac{\omega\mathcal{L}}{\mu}\|x^*\|^2 + \frac{\sigma_*^2}{\mu}\min\left(\gamma, \mathcal{O}(f(\theta) - f^*)\right)\right].$$

Clearly, there are several factors influencing the speed of the rate and the neighborhood of the convergence that require some discussion. First of all, choosing the identity map as a compression operator ($\mathcal{C}(x) = x$ for all $x \in \mathbb{R}^d$), we recover the same rate (11) as before ($\omega = 0$ in this case). Next, consider the case when the stochastic noise at the optimum vanishes ($\sigma_*^2 = 0$) and distillation is switched off ($\lambda = 0$) in (18). In this case, the convergence is still up to some neighborhood proportional to $\|x^*\|^2$ since compression is applied to the iterates. Intuitively, the neighborhood term $\mathcal{O}(\|x^*\|^2)$ corresponds to the compression noise at the optimum $x^*$ ((19) when $x = x^*$). Also note that the presence of this non-vanishing term $\mathcal{O}(\|x^*\|^2)$ and the variance restriction $\omega = \mathcal{O}(\mu/\mathcal{L})$ is consistent with the prior work [18].

So, the convergence neighborhood of iterates (18) has two terms, one from each source of randomness: compression noise/variance $\mathcal{O}(\|x^*\|^2)$ at the optimum and stochastic noise/variance $\mathcal{O}(\sigma_*^2)$ at the optimum. Therefore, in this case as well, distillation with a properly chosen weight parameter (partially) reduces the stochastic variance of sub-sampling.

# 7    Discussion and Future Work

Our work has provided a new interpretation of knowledge distillation, examining this mechanism for the first time from the point of view of optimization. Specifically, we have shown that knowledge distillation acts as a form of partial variance reduction, whose strength depends on the characteristics of the teacher model. This finding holds across several variants of distillation, such as self-distillation and distillation of compressed models, as well as across various families of objective functions.

Prior observations showed that significant capacity gap between the student and the teacher may in fact lead to poorer distillation performance [31]. To reconcile the issue of large capacity gap our results, notice that, in our case "better teacher" means better parameter (i.e., weights and biases) values, evaluated in terms of training loss. In particular, in the case of self-distillation, covered in Sections 4 and 5, the teacher and student architectures are identical, and hence they have the same capacity. In our second regime, distillation for compressed models (Section 6), we actually consider the case when the student network is a subnetwork of the teacher; we consider a sparsification compression operator that selects $k$ parameters for the student out of $d$ parameters of the teacher. Then, clearly, the teacher has a larger capacity with a capacity ratio $d/k \geq 1$. However, our result in this direction (Theorem 4) does not allow the capacity ratio to be arbitrarily large. Indeed, the constraint $\omega = \mathcal{O}(\mathcal{L}/\mu)$ on compression variance implies a constraint on capacity ratio since $\omega = d/k - 1$ for the sparsification operator. Thus, our result holds when the teacher's size is not significantly larger than the student's size, which is in line with the prior observations on large capacity gap.

As we mentioned, our Proposition 1 does not hold precisely for arbitrary deep non-linear neural networks. However, we showed that this simple model (5) of distillation gradient approximates the true distillation gradient reasonably well both empirically (see Figure 1) and analytically (see Appendix B.3). There is much more to investigate for the case of non-convex deep networks where exact tracking of teacher's impact across multiple layers of non-linearities becomes harder. We see our results as a promising first step towards a more complete understanding of the effectiveness of distillation. One interesting direction of future work would be to construct more complex models for distillation gradient and to investigate further connections with more complex variance-reduction methods, e.g. [4], which may yield even better-performing variants of KD.

## Acknowledgements

MS has received funding from the European Union's Horizon 2020 research and innovation programme under the Marie Skłodowska-Curie grant agreement No 101034413.

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

# Appendix

## Contents

# A  Basic Facts and Inequalities

To facilitate the reading of technical part of the work, here we present several standard inequalities and basic facts that we are going to use in the proofs.

- Usually we need to bound the sum of two error terms by individual errors using the following simple bound

$$\|a + b\|^2 \le 2\|a\|^2 + 2\|b\|^2. \tag{20}$$

A direct generalization of this inequality for arbitrary number of summation terms is the following one:

$$\left\| \sum_{i=1}^{n} a_i \right\|^2 \le n \sum_{i=1}^{n} \|a_i\|^2. \tag{21}$$

This inequality can be seen as a special case of Jensen's inequality for convex functions $h\colon \mathbb{R}^d \to \mathbb{R}$,

$$h\left( \sum_{i=1}^{n} \alpha_i x_i \right) \le \sum_{i=1}^{n} \alpha_i h(x_i),$$

where $x_i \in \mathbb{R}^d$ are any vectors and $\alpha_i \in [0,1]$ with $\sum_{i=1}^{n} \alpha_i = 1$. Then, (20) follows from Jensen's inequality when $h(x) = \|x\|^2$ and $\alpha_1 = \cdots = \alpha_n = \frac{1}{n}$. A more general version of the Jensen's inequality, from the perspective of probability theory, is

$$h(\mathbb{E}z) \le \mathbb{E}h(z) \tag{22}$$

for any convex function $h$ and random vector $z \in \mathbb{R}^d$.

Another extension of (20), which we actually apply in that form, is Peter-Paul inequality given by

$$\langle a, b \rangle \le \frac{1}{2s} \|a\|^2 + \frac{s}{2} \|b\|^2, \tag{23}$$

for any positive $s > 0$. Then, (20) is the special case of (23) with $s = 1$.

- Typically, a function $f$ is called $L$-smooth if its gradient is Lipschitz continuous with Lipschitz constant $L \ge 0$, namely

$$\|\nabla f(x) - \nabla f(y)\| \le L \|x - y\|, \tag{24}$$

In particular, smoothness inequality (24) implies the following quadratic upper bound [35]:

$$f(y) \le f(x) + \langle \nabla f(x), y - x \rangle + \frac{L}{2} \|y - x\|^2, \tag{25}$$

for all points $x, y \in \mathbb{R}^d$. If the function $f$ is additionally convex, then the following lower bound holds too:

$$f(y) \ge f(x) + \langle \nabla f(x), y - x \rangle + \frac{1}{2L} \|\nabla f(y) - \nabla f(x)\|^2. \tag{26}$$

- As we mentioned in the main part of the paper, function $f$ is called $\mu$-strongly convex if the following inequality holds

$$f(y) \ge f(x) + \langle \nabla f(x), y - x \rangle + \frac{\mu}{2} \|y - x\|^2, \tag{27}$$

for all points $x, y \in \mathbb{R}^d$. Recall that strong quasi-convexity assumption (1) is the special case of (27) when $y = x^*$. In other words, strong convexity (27) implies strong quasi-convexity (1). Continuing this chain of conditions, strong quasi-convexity (1) implies PL condition (9), which, in turn, implies the so-called *Quadratic Growth* condition given below

$$f(x) - f^* \ge \frac{\mu}{8} \|x - x^*\|^2, \tag{28}$$

for all $x \in \mathbb{R}^d$. Derivations of these implications and relationship with other conditions can be found in [17].

- As many analyses with linear or exponential convergence speed, our analysis also uses a very standard transformation from a single-step recurrence relation to convergence inequality. Specifically, assume we have the following recursion for $e^t$, $t = 0, 1, 2, \ldots$:

$$e^{t+1} \le (1-\eta)e^t + N,$$

with constants $\eta \in (0,1]$ and $N \ge 0$. Then, repeated application of the above recursion gives

$$e^t \le (1-\eta)^t e^0 + (1-\eta)^{t-1} N + (1-\eta)^{t-2} N + \cdots + (1-\eta)^0 N$$

$$\le (1-\eta)^t e^0 + N \sum_{j=0}^{\infty} (1-\eta)^j = (1-\eta)^t e^0 + \frac{N}{\eta}. \quad (29)$$

• If $z \in \mathbb{R}^d$ is a random vector and $x \in \mathbb{R}^d$ is fixed (or has randomness independent of $z$), then the following decomposition holds:

$$\mathbb{E}\left[\|x+z\|^2\right] = \|x\|^2 + \mathbb{E}\left[\|z\|^2\right] \quad (30)$$

# B   Proofs for Section 3

For the sake of presentation, we first consider binary logistic regression problem as a special case of multi-class classification.

## B.1   Binary Logistic Regression

In this case we have $d$-dimensional input vectors $a_n \in \mathbb{R}^d$ with their binary true labels $b_n \in \mathcal{B} = [0,1]$, predictor $\phi_x(a) = \sigma(x^\top \bar{a}) \in (0,1)$ with parameters $x \in \mathcal{P} = \mathbb{R}^{d+1}$ and lifted input vector $\bar{a} = [a\ 1]^\top \in \mathbb{R}^{d+1}$ (to avoid additional notation for the bias terms), where $\sigma(t) = \frac{1}{1+e^{-t}}$, $t \in \mathbb{R}$ is the sigmoid function. Besides, the loss is given by the cross entropy loss below

$$\ell(p,q) = H\left(\left[\begin{smallmatrix} q \\ 1-q \end{smallmatrix}\right], \left[\begin{smallmatrix} p \\ 1-p \end{smallmatrix}\right]\right) = -q \log p - (1-q)\log(1-p), \quad p,q \in [0,1].$$

Based on (2) and (3), we have

$$
\begin{aligned}
f_n(x \mid \theta, \lambda) &= (1-\lambda)f_n(x) + \lambda f_n(x \mid \theta) \\
&= (1-\lambda)\ell(\phi_x(a_n), b_n) + \lambda\ell(\phi_x(a_n), \phi_\theta(a_n)) \\
&= \ell(\phi_x(a_n), (1-\lambda)b_n + \lambda\phi_\theta(a_n)) \\
&= \ell(\sigma(x^\top \bar{a}_n), (1-\lambda)b_n + \lambda\sigma(\theta^\top \bar{a}_n)) = \ell(\sigma(x^\top \bar{a}_n), s_n) \\
&= s_n \log\left(1 + e^{-x^\top \bar{a}_n}\right) + (1-s_n)\log\left(1 + e^{x^\top \bar{a}_n}\right),
\end{aligned}
$$

where $s_n = (1-\lambda)b_n + \lambda\sigma(\theta^\top \bar{a}_n)$ are the soft labels. Notice that $f_n(x \mid \theta, \lambda = 0) = f_n(x)$. Next, we derive an expression for the stochastic gradient for the above objective, namely the gradient $\nabla_x f_n(x \mid \theta, \lambda)$ of the loss associated with $n^{th}$ data point $(a_n, b_n)$.

$$\nabla_x\left[s_n \log\left(1 + e^{-x^\top \bar{a}_n}\right)\right] = -s_n \frac{e^{-x^\top \bar{a}_n}}{1 + e^{-x^\top \bar{a}_n}} \bar{a}_n = -s_n \sigma(-x^\top \bar{a}_n)\,\bar{a}_n,$$

$$\nabla_x\left[(1-s_n) \log\left(1 + e^{x^\top \bar{a}_n}\right)\right] = (1-s_n) \frac{e^{x^\top \bar{a}_n}}{1 + e^{x^\top \bar{a}_n}} \bar{a}_n = (1-s_n)\sigma(x^\top \bar{a}_n)\,\bar{a}_n.$$

Hence, using the identity $\sigma(t) + \sigma(-t) = 1$ for the sigmoid function, we get

$$
\begin{aligned}
\nabla_x f_n(x \mid \theta, \lambda) &= \left[-s_n(1 - \sigma(x^\top \bar{a}_n)) + (1-s_n)\sigma(x^\top \bar{a}_n)\right]\bar{a}_n \\
&= \left[\sigma(x^\top \bar{a}_n) - s_n\right]\bar{a}_n \\
&= \left[\sigma(x^\top \bar{a}_n) - (1-\lambda)b_n - \lambda\sigma(\theta^\top \bar{a}_n)\right]\bar{a}_n \\
&= \left(\sigma(x^\top \bar{a}_n) - b_n\right)\bar{a}_n - \lambda\left(\sigma(\theta^\top \bar{a}_n) - b_n\right)\bar{a}_n = \nabla f_n(x) - \lambda\nabla f_n(\theta).
\end{aligned}
$$

Thus, the distillation gradient for binary logistic regression tasks the same form as (5).

## B.2 Multi-class Classification with Soft-max

Now we extend the above steps for multi-class problem. Here we consider a $K$-classification model with one hidden layer that has soft-max as the last layer, i.e., the forward pass has the following steps:

$$a_n \to X^\top a_n \to \phi_X(a_n) := \sigma(X^\top a_n) \in \mathbb{R}^K,$$

where $X = [x_1 \; x_2 \; \ldots \; x_K] \in \mathbb{R}^{d \times K}$ are the student's model parameters, $a_n \in \mathcal{A} = \mathbb{R}^d$ is the input data and $\sigma$ is the soft-max function (as a generalization of sigmoid function). Then we simplify the loss

$$
\begin{aligned}
f_n(X \mid \Theta, \lambda) &= (1-\lambda)f_n(X) + \lambda f_n(X \mid \Theta) \\
&= (1-\lambda)\ell(\phi_X(a_n), b_n) + \lambda\ell(\phi_X(a_n), \phi_\Theta(a_n)) \\
&= \ell(\phi_X(a_n), (1-\lambda)b_n + \lambda\phi_\Theta(a_n)) \\
&= \ell(\sigma(X^\top a_n), s_n) \\
&= -\sum_{k=1}^K s_{n,k} \log \sigma(X^\top a_n)_k = -\sum_{k=1}^K s_{n,k} \log \frac{e^{x_k^\top a_n}}{\sum_{j=1}^K e^{x_j^\top a_n}} \\
&= \sum_{k=1}^K s_{n,k} \left( \log \sum_{j=1}^K e^{x_j^\top a_n} - \log e^{x_k^\top a_n} \right) = \sum_{k=1}^K s_{n,k} \log \sum_{j=1}^K e^{x_j^\top a_n} - \sum_{k=1}^K s_{n,k} \log e^{x_k^\top a_n} \\
&= \log \sum_{k=1}^K e^{x_k^\top a_n} - \sum_{k=1}^K s_{n,k} \log e^{x_k^\top a_n} = \log \sum_{k=1}^K e^{x_k^\top a_n} - \sum_{k=1}^K s_{n,k} x_k^\top a_n,
\end{aligned}
$$

where $s_n = (1-\lambda)b_n + \lambda\phi_\Theta(a_n) \in \mathbb{R}^K$ are the soft labels. Next, we derive an expression for the stochastic gradient for the above objective, namely the gradient $\nabla_{x_k} f_n(X \mid \Theta, \lambda)$ of the loss associated with $n^{th}$ data point $(a_n, b_n)$.

$$
\begin{aligned}
\nabla_{x_k} f_n(X \mid \Theta, \lambda) &= \nabla_{x_k} \left[ \log \sum_{k=1}^K e^{x_k^\top a_n} - \sum_{k=1}^K s_{n,k} x_k^\top a_n \right] = \frac{e^{x_k^\top a_n}}{\sum_{i=1}^K e^{x_i^\top a_n}} a_n - s_{n,k} a_n \\
&= \left( \sigma(X^\top a_n) - s_n \right)_k a_n = \left( \sigma(X^\top a_n) - b_n \right)_k a_n - \lambda \left( \sigma(\Theta^\top a_n) - b_n \right) a_n \\
&= \nabla_{x_k} f_n(X) - \lambda \nabla_{\theta_k} f_n(\Theta).
\end{aligned}
$$

Again, we get the same expression (5) for the distillation gradient

$$\nabla_X f_n(X \mid \Theta, \lambda) = \nabla_X f_n(X) - \lambda \nabla_\Theta f_n(\Theta).$$

## B.3 Generic Non-linear Classification

Finally, consider arbitrary classification model that ends with linear layer and soft-max as the last layer, i.e., the forward pass has the following steps

$$a_n \to \psi_n(x) \to \phi_x(a_n) := \sigma(\psi_n(x)) \in \mathbb{R}^K,$$

where $a_n \in \mathcal{A} = \mathbb{R}^d$ is the input data and $\psi_n(x) \in \mathbb{R}^K$ are the logits with respect to the model parameters $x$. Denote $\varphi_n(z) := \ell(\sigma(z), b_n)$ the loss associated with logits $z$ and the true label $b_n$. In words, $\psi_n$ gives the logits from the input data, while $\varphi_n$ gives the loss from given logits. Then, clearly we have the following representation for the loss function $f_n(x) = \varphi_n(\psi_n(x))$. Next, let us

simplify the distillation loss as

$$
\begin{aligned}
f_n(x \mid \theta, \lambda) &= (1 - \lambda) f_n(x) + \lambda f_n(x \mid \theta) \\
&= (1 - \lambda) \ell(\phi_x(a_n), b_n) + \lambda \ell(\phi_x(a_n), \phi_\theta(a_n)) \\
&= \ell(\phi_x(a_n), (1 - \lambda) b_n + \lambda \phi_\theta(a_n)) \\
&= \ell(\sigma(\psi_n(x), (1 - \lambda) b_n + \lambda \sigma(\psi_n(\theta)))) \\
&= \ell(\sigma(\psi_n(x), s_n) \\
&= -\sum_{k=1}^{K} s_{n,k} \log \sigma(\psi_n(x))_k = -\sum_{k=1}^{K} s_{n,k} \log \frac{e^{\psi_{n,k}(x)}}{\sum_{j=1}^{K} e^{\psi_{n,j}(x)}} \\
&= \sum_{k=1}^{K} s_{n,k} \left( \log \sum_{j=1}^{K} e^{\psi_{n,j}(x)} - \psi_{n,k}(x) \right) \\
&= \sum_{k=1}^{K} s_{n,k} \log \sum_{j=1}^{K} e^{\psi_{n,j}(x)} - \sum_{k=1}^{K} s_{n,k} \psi_{n,k}(x) \\
&= \log \sum_{k=1}^{K} e^{\psi_{n,k}(x)} - \sum_{k=1}^{K} s_{n,k} \psi_{n,k}(x),
\end{aligned}
$$

where $s_n = (1 - \lambda) b_n + \lambda \phi_\theta(a_n)$. Now we need to differentiate obtained expression and derive an expression for the stochastic gradient for the above objective, namely the gradient $\nabla_x f_n(x \mid \theta, \lambda)$ of the loss associated with $n^{th}$ data point $(a_n, b_n)$. Applying the gradient operator, we get

$$
\begin{aligned}
\nabla_x f_n(x \mid \theta, \lambda) &= \nabla_x \left[ \log \sum_{k=1}^{K} e^{\psi_{n,k}(x)} - \sum_{k=1}^{K} s_{n,k} \psi_{n,k}(x) \right] \\
&= \sum_{k=1}^{K} \frac{e^{\psi_{n,k}(x)}}{\sum_{j=1}^{K} e^{\psi_{n,j}(x)}} \nabla_x \psi_{n,k}(x) - \sum_{k=1}^{K} s_{n,k} \nabla_x \psi_{n,k}(x) \\
&= \sum_{k=1}^{K} (\sigma(\psi_n(x)) - s_n)_k \nabla_x \psi_n(x) \\
&= J\psi_n(x) (\sigma(\psi_n(x)) - s_n),
\end{aligned}
$$

where

$$
J\psi_n(x) := \frac{\partial \psi_n(x)}{\partial x} = [\nabla \psi_{n,1}(x) \ \nabla \psi_{n,2}(x) \dots \nabla \psi_{n,K}(x)] \in \mathbb{R}^{d \times K}
$$

is the Jacobian of vector-valued function $\psi_n \colon \mathbb{R}^d \to \mathbb{R}^K$. From the derivation so far we imply that

$$
\sigma(\psi_n(x)) - s_n = (\sigma(\psi_n(x)) - b_n) - \lambda (\sigma(\psi_n(\theta)) - b_n) = \nabla \varphi_n(\psi_n(x)) - \lambda \nabla \varphi_n(\psi_n(\theta)).
$$

Taking into account that $f_n(x) = \varphi_n(\psi_n(x))$, we show the following form for the distilled gradient

$$
\begin{aligned}
\nabla_x f_n(x \mid \theta, \lambda) &= J\psi_n(x) (\nabla \varphi_n(\psi_n(x)) - \lambda \nabla \varphi_n(\psi_n(\theta))) \\
&= \frac{\partial \psi_n(x)}{\partial x} \frac{\partial f_n(x)}{\partial \psi_n(x)} - \lambda \frac{\partial \psi_n(x)}{\partial x} \frac{\partial f_n(\theta)}{\partial \psi_n(\theta)}.
\end{aligned}
$$

## C Proofs for Section 4

Before we proceed to the proofs of Theorems 1 and 2, we prove a key lemma that will be useful in both proofs. The lemma we are about to present covers *Part 2 (Optimal distillation weight)* and *Part 3 (Impact of the teacher)* of the proof overview discussed in the main content.

### C.1 Key lemma

To simplify the expressions in our proofs, let us introduce some notation describing stochastic gradients. Denote the signal-to-noise ratio with respect to parameters $\theta$ by

$$\beta(\theta) := \frac{\|\nabla f(\theta)\|^2}{\mathbb{E}\left[\|\nabla f_\xi(\theta)\|^2\right]} \in [0,1], \tag{31}$$

and correlation coefficient between stochastic gradients $\nabla f_\xi(x)$ and $\nabla f_\xi(y)$ by

$$\rho(x,y) := \frac{\mathrm{Cov}(x,y)}{\mathrm{Var}(x)\mathrm{Var}(y)} \in [-1,1], \tag{32}$$

where

$$\mathrm{Cov}(x,y) := \mathbb{E}\left[\langle \nabla f_\xi(x) - \nabla f(x), \nabla f_\xi(y) - \nabla f(y)\rangle\right], \qquad \mathrm{Var}(x) := \sqrt{\mathrm{Cov}(x,x)},$$

are the covariance and variance respectively.

**Lemma 1.** *Let $N(\lambda) = \lambda^2 \|\nabla f(\theta)\|^2 + c\gamma \mathbb{E}\left[\|\nabla f_\xi(x^*) - \lambda \nabla f_\xi(\theta)\|^2\right]$ for some constant $c \geq 0$. Then, the optimal $\lambda$ that minimizes $N(\lambda)$ is given by*

$$\lambda_* = \frac{\mathbb{E}\left[\langle \nabla f_\xi(x^*), \nabla f_\xi(\theta)\rangle\right]}{\mathbb{E}\left[\|\nabla f_\xi(\theta)\|^2\right] + \frac{1}{c\gamma}\|\nabla f(\theta)\|^2}. \tag{33}$$

*Moreover,*

$$\frac{N(\lambda_*)}{N(0)} = 1 - \rho^2(x^*,\theta)\frac{1-\beta(\theta)}{1+\frac{1}{c\gamma}\beta(\theta)} \leq \min\left(1, \mathcal{O}\left(\frac{1}{\gamma}(f(\theta) - f^*)\right)\right).$$

*Proof.* Notice that $N(\lambda)$ is quadratic in $\lambda$ and using the first-order optimality condition, we conclude

$$\frac{d}{d\lambda}N(\lambda) = 2\lambda\|\nabla f(\theta)\|^2 + c\gamma\left(-2\mathbb{E}\left[\langle \nabla f_\xi(x^*), \nabla f_\xi(\theta)\rangle\right] + 2\lambda\mathbb{E}\left[\|\nabla f_\xi(\theta)\|^2\right]\right) = 0,$$

we get (33). Furthermore, plugging the expression of $\lambda_*$ into $N(\lambda)$, we get

$$
\begin{aligned}
N(\lambda_*) &= c\gamma\left(\frac{\lambda_*^2}{c\gamma}\|\nabla f(\theta)\|^2 + \mathbb{E}\left[\|\nabla f_\xi(x^*)\|^2\right] - 2\lambda_*\mathbb{E}\left[\langle \nabla f_\xi(x^*), \nabla f_\xi(\theta)\rangle\right] + \lambda_*^2\mathbb{E}\left[\|\nabla f_\xi(\theta)\|^2\right]\right) \\
&= c\gamma\left(\mathbb{E}\left[\|\nabla f_\xi(x^*)\|^2\right] - 2\lambda_*\mathbb{E}\left[\langle \nabla f_\xi(x^*), \nabla f_\xi(\theta)\rangle\right] + \lambda_*^2\left(\mathbb{E}\left[\|\nabla f_\xi(\theta)\|^2\right] + \frac{1}{c\gamma}\|\nabla f(\theta)\|^2\right)\right) \\
&= c\gamma\left(\mathbb{E}\left[\|\nabla f_\xi(x^*)\|^2\right] - \frac{(\mathbb{E}\left[\langle \nabla f_\xi(x^*), \nabla f_\xi(\theta)\rangle\right])^2}{\mathbb{E}\left[\|\nabla f_\xi(\theta)\|^2\right] + \frac{1}{c\gamma}\|\nabla f(\theta)\|^2}\right) \\
&= c\gamma\mathbb{E}\left[\|\nabla f_\xi(x^*)\|^2\right]\left(1 - \frac{(\mathbb{E}\left[\langle \nabla f_\xi(x^*), \nabla f_\xi(\theta)\rangle\right])^2}{\left(\mathbb{E}\left[\|\nabla f_\xi(\theta)\|^2\right] + \frac{1}{c\gamma}\|\nabla f(\theta)\|^2\right)(\mathbb{E}\left[\|\nabla f_\xi(x^*)\|^2\right])}\right).
\end{aligned}
$$

Note that $N(0) = c\gamma\sigma_*^2$. From $\mathbb{E}\left[\nabla f_\xi(x^*)\right] = \nabla f(x^*) = 0$, we imply that

$$\mathrm{Cov}(x^*,\theta) = \mathbb{E}\left[\langle \nabla f_\xi(x^*), \nabla f_\xi(\theta) - \nabla f(\theta)\rangle\right] = \mathbb{E}\left[\langle \nabla f_\xi(x^*), \nabla f_\xi(\theta)\rangle\right],$$

and therefore

$$\rho(x^*,\theta) = \frac{\mathbb{E}\left[\langle \nabla f_\xi(x^*), \nabla f_\xi(\theta)\rangle\right]}{\sqrt{\mathbb{E}\left[\|\nabla f_\xi(x^*)\|^2\right]}\sqrt{\mathbb{E}\left[\|\nabla f_\xi(\theta)\|^2\right] - \|\nabla f(\theta)\|^2}}.$$

Now we can simplify the expression for $N(\lambda_*)$ as follows

$$
\begin{aligned}
\frac{N(\lambda_*)}{N(0)} &= 1 - \frac{(\mathbb{E}\left[\langle \nabla f_\xi(x^*), \nabla f_\xi(\theta)\rangle\right])^2}{\left(\mathbb{E}\left[\|\nabla f_\xi(\theta)\|^2\right] + \frac{1}{c\gamma}\|\nabla f(\theta)\|^2\right)\left(\mathbb{E}\left[\|\nabla f_\xi(x^*)\|^2\right]\right)} \\
&= 1 - \frac{(\mathbb{E}\left[\langle \nabla f_\xi(x^*), \nabla f_\xi(\theta)\rangle\right])^2}{(\mathbb{E}\left[\|\nabla f_\xi(\theta)\|^2\right] - \|\nabla f(\theta)\|^2)\left(\mathbb{E}\left[\|\nabla f_\xi(x^*)\|^2\right]\right)}\frac{\mathbb{E}\left[\|\nabla f_\xi(\theta)\|^2\right] - \|\nabla f(\theta)\|^2}{\mathbb{E}\left[\|\nabla f_\xi(\theta)\|^2\right] + \frac{1}{c\gamma}\|\nabla f(\theta)\|^2} \\
&= 1 - \rho^2(x^*,\theta)\frac{1-\beta(\theta)}{1+\frac{1}{c\gamma}\beta(\theta)}.
\end{aligned}
$$

Here, $\rho(x^*,\theta)$ is the correlation coefficient between stochastic gradients at $x^*$ and $\theta$. Hence, we showed with tuned distillation weight the neighborhood can shrink by some factor depending on the teacher's parameters. In the extreme case when the teacher $\theta = x^*$ is optimal, we have $\rho(x^*,\theta) = 1$, $\beta(\theta) = 0$ and, thus, no neighborhood $N(\lambda_*) = 0$. This hints us on the fact that the reduction factor $N(\lambda_*)/N(0)$ of the neighborhood is controlled by the "quality" of the teacher.

To make this argument rigorous, consider the teacher's model to be away from the optimal solution $x^*$ within the limit described by the following inequality

$$
f(\theta) - f^* \leq \frac{\sigma(x^*)\sigma(\theta)}{\mathcal{L}}, \tag{34}
$$

where $\sigma^2(x) := \mathbb{E}\left[\|\nabla f_\xi(x)\|^2\right]$ is the second moment of the stochastic gradients. Without loss of generality, we assume that $\sigma^2(x) > 0$ for all parameter choices $x \in \mathbb{R}^d$: otherwise we have $\sigma_*^2 = 0$ and even plain SGD ensures full variance reduction. Then, we can simplify the reduction factor as

$$
\begin{aligned}
1 - \rho^2(x^*,\theta)\frac{1-\beta(\theta)}{1+\frac{1}{c\gamma}\beta(\theta)} &= 1 - \frac{(\mathbb{E}\left[\langle \nabla f_\xi(x^*), \nabla f_\xi(\theta)\rangle\right])^2}{\mathbb{E}\left[\|\nabla f_\xi(\theta)\|^2\right]\mathbb{E}\left[\|\nabla f_\xi(x^*)\|^2\right]}\frac{1}{1+\frac{1}{c\gamma}\beta(\theta)} \\
&= 1 - \left(\frac{\sigma^2(\theta) + \sigma^2(x^*) - \mathbb{E}\left[\|\nabla f_\xi(x^*) - \nabla f_\xi(\theta)\|^2\right]}{2\sigma(\theta)\sigma(x^*)}\right)^2\frac{1}{1+\frac{1}{c\gamma}\beta(\theta)} \\
&= 1 - \left(\frac{\sigma^2(\theta) + \sigma^2(x^*)}{2\sigma(\theta)\sigma(x^*)} - \frac{\mathbb{E}\left[\|\nabla f_\xi(x^*) - \nabla f_\xi(\theta)\|^2\right]}{2\sigma(\theta)\sigma(x^*)}\right)^2\frac{1}{1+\frac{1}{c\gamma}\beta(\theta)} \\
&\overset{(10)+(34)}{\leq} 1 - \left(1 - \frac{\mathcal{L}(f(\theta) - f^*)}{\sigma(\theta)\sigma(x^*)}\right)^2\frac{1}{1+\frac{2L}{c\gamma}\frac{f(\theta)-f^*}{\sigma^2(\theta)}} \\
&\leq \left(2 + \frac{2L}{c\gamma\mathcal{L}}\frac{\sigma(x^*)}{\sigma(\theta)}\right)\frac{\mathcal{L}(f(\theta) - f^*)}{\sigma(\theta)\sigma(x^*)} \\
&= \left(\frac{2\mathcal{L}}{\sigma(x^*)\sigma(\theta)} + \frac{1}{c\gamma}\frac{2L}{\sigma^2(\theta)}\right)(f(\theta) - f^*) = \mathcal{O}\left(\frac{1}{\gamma}(f(\theta) - f^*)\right),
\end{aligned}
$$

where the last inequality used $1 - \frac{(1-u)^2}{1+uv} \leq (2+v)u$ for all $u$, $v \geq 0$. □

## C.2 Proof of Theorem 1

Denote by $\mathbb{E}_t\left[\cdot\right] := \mathbb{E}\left[\cdot \mid x^t\right]$ the conditional expectation with respect to $x^t$. Then, we start bounding the error using the update rule (7).

$$\mathbb{E}_t\left[\|x^{t+1}-x^*\|^2\right]$$

$$= \|x^t-x^*\|^2 - 2\gamma\left\langle x^t-x^*, \nabla f(x^t) - \lambda\nabla f(\theta)\right\rangle + \gamma^2\mathbb{E}_t\left[\|\nabla f_\xi(x^t) - \lambda\nabla f_\xi(\theta)\|^2\right]$$

$$= \|x^t-x^*\|^2 - 2\gamma\left\langle x^t-x^*, \nabla f(x^t)\right\rangle + 2\gamma\lambda\left\langle x^t-x^*, \nabla f(\theta)\right\rangle + \gamma^2\mathbb{E}_t\left[\|\nabla f_\xi(x^t) - \lambda\nabla f_\xi(\theta)\|^2\right]$$

$$\overset{(1)+(20)}{\leq} (1-\gamma\mu)\|x^t-x^*\|^2 - 2\gamma(f(x^t)-f(x^*)) + 2\gamma\lambda\left\langle x^t-x^*, \nabla f(\theta)\right\rangle$$
$$+ 2\gamma^2\mathbb{E}_t\left[\|\nabla f_\xi(x^t) - \nabla f_\xi(x^*)\|^2\right] + 2\gamma^2\mathbb{E}\left[\|\nabla f_\xi(x^*) - \lambda\nabla f_\xi(\theta)\|^2\right]$$

$$\overset{(28)+(23)}{\leq} (1-\gamma\mu)\|x^t-x^*\|^2 - \gamma(f(x^t)-f(x^*)) - \frac{\gamma\mu}{8}\|x^t-x^*\|^2 + \frac{\gamma\mu}{8}\|x^t-x^*\|^2 + \frac{8\gamma}{\mu}\lambda^2\|\nabla f(\theta)\|^2$$
$$+ 2\gamma^2\mathbb{E}_t\left[\|\nabla f_\xi(x^t) - \nabla f_\xi(x^*)\|^2\right] + 2\gamma^2\mathbb{E}\left[\|\nabla f_\xi(x^*) - \lambda\nabla f_\xi(\theta)\|^2\right]$$

$$\overset{(3)}{\leq} (1-\gamma\mu)\|x^t-x^*\|^2 - \gamma(f(x^t)-f(x^*)) + \frac{8\gamma}{\mu}\lambda^2\|\nabla f(\theta)\|^2$$
$$+ 4\gamma^2\mathcal{L}(f(x^t)-f(x^*)) + 2\gamma^2\mathbb{E}\left[\|\nabla f_\xi(x^*) - \lambda\nabla f_\xi(\theta)\|^2\right]$$

$$= (1-\gamma\mu)\|x^t-x^*\|^2 - \gamma\left(1-4\gamma\mathcal{L}\right)(f(x^t)-f(x^*))$$
$$+ \frac{8\gamma}{\mu}\lambda^2\|\nabla f(\theta)\|^2 + 2\gamma^2\mathbb{E}\left[\|\nabla f_\xi(x^*) - \lambda\nabla f_\xi(\theta)\|^2\right]$$

$$\leq (1-\gamma\mu)\|x^t-x^*\|^2 + \frac{8\gamma}{\mu}\lambda^2\|\nabla f(\theta)\|^2 + 2\gamma^2\mathbb{E}\left[\|\nabla f_\xi(x^*) - \lambda\nabla f_\xi(\theta)\|^2\right],$$

where we used Peter-Paul inequality (23) with parameter $s = \frac{8}{\mu}$ and the step-size bound $\gamma \leq \frac{1}{4\mathcal{L}}$ in the last inequality. Applying full expectation and unrolling the recursion, we get

$$\mathbb{E}\left[\|x^t-x^*\|^2\right] \leq (1-\gamma\mu)\mathbb{E}\left[\|x^{t-1}-x^*\|^2\right] + \frac{8\gamma}{\mu}\lambda^2\|\nabla f(\theta)\|^2 + 2\gamma^2\mathbb{E}\left[\|\nabla f_\xi(x^*) - \lambda\nabla f_\xi(\theta)\|^2\right]$$

$$\overset{(29)}{\leq} (1-\gamma\mu)^t\|x^0-x^*\|^2 + \frac{8\lambda^2}{\mu^2}\|\nabla f(\theta)\|^2 + \frac{2\gamma}{\mu}\mathbb{E}\left[\|\nabla f_\xi(x^*) - \lambda\nabla f_\xi(\theta)\|^2\right]$$

$$= (1-\gamma\mu)^t\|x^0-x^*\|^2 + \frac{8}{\mu^2}N_1(\lambda), \tag{35}$$

where $N_1(\lambda) := \lambda^2\|\nabla f(\theta)\|^2 + \frac{\gamma\mu}{4}\mathbb{E}\left[\|\nabla f_\xi(x^*) - \lambda\nabla f_\xi(\theta)\|^2\right]$. Applying Lemma 1 with $c = \mu/4$, we imply that for some $\lambda = \lambda_*$ the neighborhood size is

$$N_1(\lambda_*) \overset{\text{Lemma 1}}{\leq} N_1(0) \cdot \min\left(1, \mathcal{O}\left(\frac{1}{\gamma}(f(\theta)-f^*)\right)\right) = \frac{\mu\sigma_*^2}{4} \cdot \min\left(\gamma, \mathcal{O}(f(\theta)-f^*)\right).$$

Plugging the above bound of $N_1$ into (35) completes the proof.

## C.3 Proof of Theorem 2

We start the recursion from the $L$-smoothness condition of $f$. As before, $\mathbb{E}_t$ denotes conditional expectation with respect to $x^t$.

$$\mathbb{E}_t\left[f(x^{t+1}) - f^*\right]$$

$$\overset{(25)}{\leq} \left(f(x^t) - f^*\right) - \gamma\left\langle\nabla f(x^t), \nabla f(x^t) - \lambda\nabla f(\theta)\right\rangle + \frac{L\gamma^2}{2}\mathbb{E}_t\left[\|\nabla f_\xi(x^t) - \lambda\nabla f_\xi(\theta)\|^2\right]$$

$$\overset{(20)}{\leq} \left(f(x^t) - f^*\right) - \gamma\|\nabla f(x^t)\|^2 + \gamma\lambda\left\langle\nabla f(x^t), \nabla f(\theta)\right\rangle$$
$$+ L\gamma^2\mathbb{E}_t\left[\|\nabla f_\xi(x^t) - \nabla f_\xi(x^*)\|^2\right] + L\gamma^2\mathbb{E}\left[\|\nabla f_\xi(x^*) - \lambda\nabla f_\xi(\theta)\|^2\right]$$

$$\overset{(9)+(23)}{\leq} (1 - \gamma\mu)\left(f(x^t) - f^*\right) - \frac{\gamma}{4}\|\nabla f(x^t)\|^2 - \frac{\gamma\mu}{2}\left(f(x^t) - f^*\right) + \frac{\gamma}{4}\|\nabla f(x^t)\|^2 + \gamma\lambda^2\|\nabla f(\theta)\|^2$$
$$+ L\gamma^2\mathbb{E}_t\left[\|\nabla f_\xi(x^t) - \nabla f_\xi(x^*)\|^2\right] + L\gamma^2\mathbb{E}_t\left[\|\nabla f_\xi(x^*) - \lambda\nabla f_\xi(\theta)\|^2\right]$$

$$\overset{(10)}{\leq} (1 - \gamma\mu)\left(f(x^t) - f^*\right) - \frac{\gamma\mu}{2}\left(f(x^t) - f^*\right) + \gamma\lambda^2\|\nabla f(\theta)\|^2$$
$$+ 2L\mathcal{L}\gamma^2\left(f(x^t) - f^*\right) + L\gamma^2\mathbb{E}_t\left[\|\nabla f_\xi(x^*) - \lambda\nabla f_\xi(\theta)\|^2\right]$$

$$= (1 - \gamma\mu)\left(f(x^t) - f^*\right) - \frac{\gamma\mu}{2}\left(1 - \gamma\frac{4L\mathcal{L}}{\mu}\right)\left(f(x^t) - f^*\right)$$
$$+ \gamma\lambda^2\|\nabla f(\theta)\|^2 + L\gamma^2\mathbb{E}\left[\|\nabla f_\xi(x^*) - \lambda\nabla f_\xi(\theta)\|^2\right]$$

$$\leq (1 - \gamma\mu)\left(f(x^t) - f^*\right) + \gamma\lambda^2\|\nabla f(\theta)\|^2 + L\gamma^2\mathbb{E}\left[\|\nabla f_\xi(x^*) - \lambda\nabla f_\xi(\theta)\|^2\right],$$

where in the last inequality we used step-size bound $\gamma \leq \frac{1}{4\mathcal{L}}\frac{\mu}{L}$. Applying full expectation and unrolling the recursion, we get

$$\mathbb{E}\left[f(x^t) - f^*\right] \leq (1 - \gamma\mu)\mathbb{E}\left[f(x^{t-1}) - f^*\|^2\right] + \gamma\lambda^2\|\nabla f(\theta)\|^2 + L\gamma^2\mathbb{E}\left[\|\nabla f_\xi(x^*) - \lambda\nabla f_\xi(\theta)\|^2\right]$$

$$\overset{(29)}{\leq} (1 - \gamma\mu)^t\left(f(x^0) - f^*\right) + \frac{\lambda^2}{\mu}\|\nabla f(\theta)\|^2 + \frac{L\gamma}{\mu}\mathbb{E}\left[\|\nabla f_\xi(x^*) - \lambda\nabla f_\xi(\theta)\|^2\right]$$

$$= (1 - \gamma\mu)^t\left(f(x^0) - f^*\right) + \frac{1}{\mu}N_2(\lambda), \tag{36}$$

where $N_2(\lambda) := \lambda^2\|\nabla f(\theta)\|^2 + L\gamma\mathbb{E}\left[\|\nabla f_\xi(x^*) - \lambda\nabla f_\xi(\theta)\|^2\right]$. Similar to the previous case, we applying Lemma 1 with $c = L$ and conclude that for some $\lambda = \lambda_*$ the neighborhood size is

$$N_2(\lambda_*) \overset{\text{Lemma 1}}{\leq} N_2(0)\cdot\min\left(1, \mathcal{O}\left(\frac{1}{\gamma}(f(\theta) - f^*)\right)\right) = L\sigma_*^2\cdot\min\left(\gamma, \mathcal{O}(f(\theta) - f^*)\right).$$

Plugging the above bound of $N_2$ into (36) completes the proof.

# D Proofs for Section 5

## D.1 Proof of Theorem 3

Again, we start the recursion from the smoothness condition of $f$.

$$\mathbb{E}_t\left[f(x^{t+1}) - f^*\right]$$

$$\overset{(25)}{\leq} \left(f(x^t) - f^*\right) - \gamma\left\langle\nabla f(x^t), \nabla f(x^t)\right\rangle + \frac{L\gamma^2}{2}\mathbb{E}_t\left[\|\nabla f_\xi(x^t) - \nabla f_\xi(\theta) + \nabla f(\theta)\|^2\right]$$

$$\overset{(21)}{\leq} \left(f(x^t) - f^*\right) - \gamma\|\nabla f(x^t)\|^2$$
$$+ \frac{3}{2}L\gamma^2\mathbb{E}_t\left[\|\nabla f_\xi(x^t) - \nabla f_\xi(x^*)\|^2\right] + \frac{3}{2}L\gamma^2\mathbb{E}\left[\|\nabla f_\xi(\theta) - \nabla f_\xi(x^*)\|^2\right] + \frac{3}{2}L\gamma^2\|\nabla f(\theta)\|^2$$

$$\overset{(9)+(10)}{\leq} (1 - \gamma\mu)\left(f(x^t) - f^*\right) - \gamma\mu\left(f(x^t) - f^*\right)$$
$$+ 3L\mathcal{L}\gamma^2\left(f(x^t) - f^*\right) + 3L\mathcal{L}\gamma^2\left(f(\theta) - f^*\right) + 3L^2\gamma^2(f(\theta) - f^*)$$

$$\leq (1 - \gamma\mu)\left(f(x^t) - f^*\right) + 3L(L + \mathcal{L})\gamma^2\left(f(\theta) - f^*\right),$$

where we used step-size bound $\gamma \leq \frac{\mu}{3L\mathcal{L}}$ in the last step. Therefore,

$$
\begin{aligned}
\mathbb{E}\left[f(x^t) - f^*\right] &\leq (1 - \gamma\mu)\mathbb{E}\left[f(x^t) - f^*\right] + 3L(L + \mathcal{L})\gamma^2 \left(f(\theta) - f^*\right) \\
&\stackrel{(29)}{\leq} (1 - \gamma\mu)^t(f(x^0) - f^*) + \frac{3L(L + \mathcal{L})}{\mu} \cdot \gamma \left(f(\theta) - f^*\right),
\end{aligned}
$$

which concludes the proof.

# E  Proofs for Section 6

First, we break the update rule (18) into two parts by introducing an auxiliary model parameters $y^t \in \mathbb{R}^d$:

$$
\begin{aligned}
y^{t+1} &= x^t - \gamma(\nabla f_\xi(x^t) - \lambda\nabla f_\xi(\theta)), \\
x^{t+1} &= \mathcal{C}(y^{t+1}).
\end{aligned}
$$

Without loss of generality, we assume that initialization satisfies $x^0 = y^0 = \mathcal{C}(y^0)$. Then, for all $t \geq 0$ we have $x^t = \mathcal{C}(y^t)$ and the update rule of $y^{t+1}$ can be written recursively without $x^t$ via

$$
y^{t+1} = \mathcal{C}(y^t) - \gamma(\nabla f_\xi(\mathcal{C}(y^t)) - \lambda\nabla f_\xi(\theta)). \tag{37}
$$

Using unbiasedness of the compression operator $\mathcal{C}$, we decompose the error

$$
\begin{aligned}
\mathbb{E}\left[\|x^t - x^*\|^2\right] &\stackrel{(30)}{=} \mathbb{E}\left[\|x^t - y^t\|^2\right] + \mathbb{E}\left[\|y^t - x^*\|^2\right] \\
&= \mathbb{E}\left[\|\mathcal{C}(y^t) - y^t\|^2\right] + \mathbb{E}\left[\|y^t - x^*\|^2\right] \\
&\stackrel{(19)}{\leq} \omega\mathbb{E}\left[\|y^t\|^2\right] + \mathbb{E}\left[\|y^t - x^*\|^2\right] \\
&\stackrel{(20)}{\leq} (2\omega + 1)\mathbb{E}\left[\|y^t - x^*\|^2\right] + 2\omega \|x^*\|^2. \tag{38}
\end{aligned}
$$

Thus, our goal would be to analyze iterates (37) and derive the rate for $x^t$. In fact, the special case of (37) was analyzed by [18] in the non-stochastic case and without distillation ($\lambda = 0$). The analysis we provide here for (37) is based on [18], and from this perspective, our analysis can be seen as an extension of their analysis.

To avoid another notation for the expected smoothness constant (see Assumption 3), analogous to (24) we assume that $\mathcal{L}$ also satisfies the following smoothness inequality:

$$
\mathbb{E}\left[\|\nabla f_\xi(x) - \nabla f_\xi(y)\|^2\right] \leq \mathcal{L}^2\|x - y\|^2. \tag{39}
$$

## E.1  Five Lemmas

First, we need to upper bound the compression error by function suboptimality. Denote $\delta(x) := \mathcal{C}(x) - x$. Let $\mathbb{E}_\delta$ and $\mathbb{E}_\xi$ be the expectations with respect to the compression operator $\mathcal{C}$ and sampling $\xi$ respectively.

**Lemma 2** (Lemma 1 in [18]). *Let $\alpha = \frac{2\omega}{\mu}$ and $\nu = 2\omega\|x^*\|^2$. For all $x \in \mathbb{R}^d$ we have*

$$
\mathbb{E}_\delta \|\delta(x)\|^2 \leq 2\alpha \left(f(x) - f(x^*)\right) + \nu, \tag{40}
$$

*Proof.* From (20) we imply $\|x\|^2 \leq 2\|x - x^*\|^2 + 2\|x_*\|^2$, and from $\mu$-strong convexity condition (27) we get $\|x - x^*\|^2 \leq \frac{2}{\mu}\left(f(x) - f(x^*)\right)$. Putting these inequalities together, we arrive at

$$
\mathbb{E}_\delta \|\delta(x)\|^2 \leq \omega\|x\|^2 \leq 2\omega\|x - x^*\|^2 + 2\omega\|x^*\|^2 \leq \frac{4\omega}{\mu}\left(f(x) - f(x^*) + 2\omega\|x^*\|^2\right.
$$

$\square$

**Lemma 3.** *For all $x, y \in \mathbb{R}^d$ we have*

$$\mathbb{E}\|\nabla f_\xi(x + \delta(x)) - \nabla f_\xi(y)\|^2 \leq \mathcal{L}^2 \left( \|x - y\|^2 + \mathbb{E}\|\delta(x)\|^2 \right), \tag{41}$$

$$f(x) \leq \mathbb{E} f(x + \delta(x)) \leq f(y) + \langle \nabla f(y), x - y \rangle + \frac{L}{2} \|x - y\|^2 + \frac{L}{2} \mathbb{E}\|\delta(x)\|^2. \tag{42}$$

*Proof.* Fix $x$ and let $\delta = \delta(x)$. Inequality (41) follows from Lipschitz continuity of the gradient, applying expectation and using (19):

$$\mathbb{E}\|\nabla f_\xi(x + \delta) - \nabla f_\xi(y)\|^2 \stackrel{(39)}{\leq} \mathcal{L}^2 \mathbb{E}_\delta \|x + \delta - y\|^2 \stackrel{(19)+(30)}{=} \mathcal{L}^2 \left( \|x - y\|^2 + \mathbb{E}_\delta \|\delta\|^2 \right).$$

The first inequality in (42) follows by applying Jensen's inequality (22) and using (19). Since $f$ is $L$–smooth, we have

$$\mathbb{E} f(x + \delta) \stackrel{(25)}{\leq} \mathbb{E} f(y) + \langle \nabla f(y), x + \delta - y \rangle + \frac{L}{2} \|x + \delta - y\|^2$$

$$\stackrel{(19)+(30)}{=} f(y) + \langle \nabla f(y), x - y \rangle + \frac{L}{2} \|x - y\|^2 + \frac{L}{2} \mathbb{E}\|\delta\|^2.$$

$\square$

**Lemma 4.** *For all $x, y \in \mathbb{R}^d$ it holds*

$$\mathbb{E}_\delta \left\| \frac{\delta(x)}{\gamma} - \nabla f_\xi(x + \delta(x)) + \lambda \nabla f_\xi(\theta) \right\|^2$$

$$\leq 2 \|\nabla f_\xi(y) - \lambda \nabla f_\xi(\theta)\|^2 + 2\mathcal{L}^2 \|x - y\|^2 + 2 \left( \mathcal{L}^2 + \frac{1}{\gamma^2} \right) \mathbb{E}_\delta \|\delta(x)\|^2. \tag{43}$$

*Proof.* Fix $x$, and let $\delta = \delta(x)$. Then for every $y \in \mathbb{R}^d$ we can write

$$\mathbb{E}_\delta \left\| \frac{\delta}{\gamma} - \nabla f_\xi(x + \delta) + \lambda \nabla f_\xi(\theta) \right\|^2$$

$$= \mathbb{E}_\delta \left\| \frac{\delta}{\gamma} \pm \nabla f_\xi(y) - \nabla f_\xi(x + \delta) + \lambda \nabla f_\xi(\theta) \right\|^2$$

$$\stackrel{(20)}{\leq} 2\mathbb{E}_\delta \left\| \frac{\delta}{\gamma} - \nabla f_\xi(y) + \lambda \nabla f_\xi(\theta) \right\|^2 + 2\mathbb{E}_\delta \|\nabla f_\xi(y) - \nabla f_\xi(x + \delta)\|^2$$

$$\stackrel{(41)}{\leq} \frac{2}{\gamma^2} \mathbb{E}_\delta \|\delta\|^2 - \frac{2}{\gamma} \mathbb{E}_\delta \langle \delta, \nabla f_\xi(y) - \lambda \nabla f_\xi(\theta) \rangle + \|\nabla f_\xi(y) - \lambda \nabla f_\xi(\theta)\|^2 + 2\mathcal{L}^2 \left( \|x - y\|^2 + \mathbb{E}_\delta \|\delta\|^2 \right)$$

$$\stackrel{(19)}{=} \frac{2}{\gamma^2} \mathbb{E}_\delta \|\delta\|^2 + 2 \|\nabla f_\xi(y) - \lambda \nabla f_\xi(\theta)\|^2 + 2\mathcal{L}^2 \left( \|x - y\|^2 + \mathbb{E}_\delta \|\delta\|^2 \right).$$

$\square$

The next lemma generalizes the strong convexity inequality (27) (special case of $\delta(x) \equiv 0$).

**Lemma 5.** *If $f$ is L-smooth and $\mu$-strongly convex, then for all $x, y \in \mathbb{R}^d$, it holds*

$$f(y) \geq f(x) + \langle \mathbb{E}[\nabla f(x + \delta)], y - x \rangle + \frac{\mu}{2} \|y - x\|^2 - \frac{L - \mu}{2} \mathbb{E}\|\delta(x)\|^2. \tag{44}$$

*Proof.* Fix $x$ and let $\delta = \delta(x)$. Using (27) with $x \leftarrow x + \delta$, we get

$$f(y) \geq f(x + \delta) + \langle \nabla f(x + \delta), y - x - \delta \rangle + \frac{\mu}{2} \|y - x - \delta\|^2.$$

Applying expectation and (30), we get

$$f(y) \geq \mathbb{E} f(x + \delta) + \mathbb{E} \langle \nabla f(x + \delta), y - x \rangle - \mathbb{E} \langle \nabla f(x + \delta), \delta \rangle + \frac{\mu}{2} \|y - x\|^2 + \frac{\mu}{2} \mathbb{E}\|\delta\|^2.$$

It remains to bound the term $-\mathbb{E}\langle \nabla f(x + \delta), \delta \rangle$, which can be done using $L$-smoothness and applying expectation as follows:

$$-\mathbb{E}\langle \nabla f(x+\delta), \delta \rangle \overset{(25)}{\geq} \mathbb{E}\left[ f(x) - f(x+\delta) - \frac{L}{2}\|\delta\|^2 \right] = f(x) - \mathbb{E}f(x+\delta) - \frac{L}{2}\mathbb{E}\|\delta\|^2.$$

$\square$

**Lemma 6.** *Denote* $A = 2\mathcal{L} + \left(\mathcal{L}^2 + \frac{1}{\gamma^2}\right)\alpha$ *and* $B = \left(\mathcal{L} + \frac{1}{\gamma^2}\right)\nu$, *where* $\alpha, \nu$ *are defined in Lemma 2. Then*

$$\mathbb{E}\left\| \frac{\delta(x)}{\gamma} - \nabla f_\xi(x + \delta(x)) + \lambda \nabla f_\xi(\theta) \right\|^2$$
$$\leq 4A(f(x) - f(x_*)) + 2B + 4\mathbb{E}\|\nabla f_\xi(x^*) - \lambda \nabla f_\xi(\theta)\|^2, \quad (45)$$

*Proof.* Using (43) with $y = x$, we get

$$\mathbb{E}\left\| \frac{\delta(x)}{\gamma} - \nabla f_\xi(x + \delta(x)) + \lambda \nabla f_\xi(\theta) \right\|^2$$

$$\overset{(43)}{\leq} 2\mathbb{E}\|\nabla f_\xi(x) - \lambda \nabla f_\xi(\theta)\|^2 + 2\left(\mathcal{L}^2 + \frac{1}{\gamma^2}\right)\mathbb{E}\|\delta(x)\|^2$$

$$\overset{(20)+(26)+(40)}{\leq} 8\mathcal{L}(f(x) - f(x_*)) + 4\mathbb{E}\|\nabla f_\xi(x^*) - \lambda \nabla f_\xi(\theta)\|^2 + 2\left(\mathcal{L}^2 + \frac{1}{\gamma^2}\right)(2\alpha(f(x) - f(x_*)) + \nu)$$

$$= 4\left(2\mathcal{L} + \left(\mathcal{L}^2 + \frac{1}{\gamma^2}\right)\alpha\right)(f(x) - f(x_*)) + 2\left(\mathcal{L}^2 + \frac{1}{\gamma^2}\right)\nu + 4\mathbb{E}\|\nabla f_\xi(x^*) - \lambda \nabla f_\xi(\theta)\|^2.$$

$\square$

### E.2  Proof of Theorem 4

Denoting $\delta^t = \delta(y^t)$ we have $\mathcal{C}(y^t) = y^t + \delta^t)$. Then

$$\|y^{t+1} - x^*\|^2$$
$$= \|\mathcal{C}(y^t) - \gamma \nabla f_\xi(\mathcal{C}(y_t)) + \gamma \lambda \nabla f_\xi(\theta) - x^*\|^2$$
$$= \|y^t - x^* + \delta^t - \gamma \nabla f_\xi(y^t + \delta^t) + \gamma \lambda \nabla f_\xi(\theta)\|^2$$
$$= \|y^t - x^*\|^2 + 2\langle \delta^t - \gamma \nabla f_\xi(y^t + \delta^t) + \gamma \lambda \nabla f_\xi(\theta), y^t - x^* \rangle + \|\delta^t - \gamma \nabla f_\xi(y^t + \delta^t) + \gamma \lambda \nabla f_\xi(\theta)\|^2.$$

Taking conditional expectation $\mathbb{E}_t := \mathbb{E}\left[\cdot \mid y^t\right]$, we get

$$\mathbb{E}_t \left[\left\|y^{t+1} - x^*\right\|^2\right]$$

$$= \left\|y^t - x^*\right\|^2 + 2\gamma\langle \mathbb{E}_t\left[\nabla f(y^t + \delta^t)\right] - \lambda\nabla f(\theta), x^* - y^t\rangle + \mathbb{E}_t\left[\left\|\delta^t - \gamma\nabla f_\xi(y^t + \delta^t) + \gamma\lambda\nabla f_\xi(\theta)\right\|^2\right]$$

$$\overset{(44)}{\leq} \left\|y^t - x^*\right\|^2 + 2\gamma\left[f(x^*) - f(y^t) - \frac{\mu}{2}\left\|y^t - x^*\right\|^2 + \frac{L-\mu}{2}\mathbb{E}_t\left\|\delta^t\right\|^2\right] + 2\gamma\lambda\langle\nabla f(\theta), y^t - x^*\rangle$$

$$+ \gamma^2\mathbb{E}_t\left\|\frac{\delta^t}{\gamma} - \nabla f_\xi(y^t + \delta^t) + \lambda\nabla f_\xi(\theta)\right\|^2$$

$$= (1 - \gamma\mu)\left\|y^t - x^*\right\|^2 - 2\gamma(f(y^t) - f(x^*)) + \gamma(L-\mu)\mathbb{E}_t\left\|\delta^t\right\|^2 + 2\gamma\lambda\langle\nabla f(\theta), y^t - x^*\rangle$$

$$+ \gamma^2\mathbb{E}_t\left\|\frac{\delta^t}{\gamma} - \nabla f_\xi(y^t + \delta^t) + \lambda\nabla f_\xi(\theta)\right\|^2$$

$$\overset{(27)+(23)}{\leq} (1 - \gamma\mu)\left\|y^t - x^*\right\|^2 - \gamma(f(y^t) - f(x^*)) - \frac{\gamma\mu}{2}\left\|y^t - x^*\right\|^2 + \gamma(L-\mu)\mathbb{E}_t\left\|\delta^t\right\|^2$$

$$+ \frac{\gamma\mu}{2}\left\|y^t - x^*\right\|^2 + \frac{2\gamma\lambda^2}{\mu}\left\|\nabla f(\theta)\right\|^2 + \gamma^2\mathbb{E}_t\left\|\frac{\delta^t}{\gamma} - \nabla f_\xi(y^t + \delta^t) + \lambda\nabla f_\xi(\theta)\right\|^2$$

$$\overset{(45)}{\leq} (1 - \gamma\mu)\left\|y^t - x^*\right\|^2 - \gamma(f(y^t) - f(x^*)) + \gamma(L-\mu)\mathbb{E}_t\left\|\delta^t\right\|^2 + \frac{2\gamma\lambda^2}{\mu}\left\|\nabla f(\theta)\right\|^2$$

$$+ 4\gamma^2 A(f(y^t) - f(x^*)) + 2\gamma^2 B + 4\gamma^2\mathbb{E}\|\nabla f_\xi(x^*) - \lambda\nabla f_\xi(\theta)\|^2$$

$$= (1 - \gamma\mu)\left\|y^t - x^*\right\|^2 + \gamma(4\gamma A - 1)(f(y^t) - f(x^*)) + 2\gamma^2 B + \gamma(L-\mu)\mathbb{E}_t\left\|\delta^t\right\|^2$$

$$+ \frac{2\gamma\lambda^2}{\mu}\left\|\nabla f(\theta)\right\|^2 + 4\gamma^2\mathbb{E}\|\nabla f_\xi(x^*) - \lambda\nabla f_\xi(\theta)\|^2$$

$$\overset{(40)}{\leq} (1 - \gamma\mu)\left\|y^t - x^*\right\|^2 + \gamma(4\gamma A - 1)(f(y^t) - f(x^*)) + 2\gamma^2 B$$

$$+ \gamma(L-\mu)\left(2\alpha(f(x_k) - f(x_*)) + \nu\right)$$

$$+ \frac{2\gamma\lambda^2}{\mu}\left\|\nabla f(\theta)\right\|^2 + 4\gamma^2\mathbb{E}\|\nabla f_\xi(x^*) - \lambda\nabla f_\xi(\theta)\|^2$$

$$= (1 - \gamma\mu)\left\|y^t - x^*\right\|^2 + \gamma(4\gamma A + 2\alpha(L-\mu) - 1)(f(y^t) - f(x^*)) + 2\gamma^2 B + \gamma(L-\mu)\nu$$

$$+ \frac{2\gamma\lambda^2}{\mu}\left\|\nabla f(\theta)\right\|^2 + 4\gamma^2\mathbb{E}\|\nabla f_\xi(x^*) - \lambda\nabla f_\xi(\theta)\|^2,$$

where $\alpha$ and $\nu$ are as in Lemma 2 and $A$ and $B$ are defined Lemma 6. Next, we show that the bounds on $\gamma$ and $\omega$ lead to $2\gamma A + \alpha(L-\mu) \leq 1/2$. Plugging the expression of $A$, we the former inequality becomes

$$2\gamma\left(2\mathcal{L} + \left(\mathcal{L}^2 + \frac{1}{\gamma^2}\right)\alpha\right) + \alpha(L-\mu) \leq \frac{1}{2}.$$

Rearranging terms, we get

$$\frac{2\omega}{\mu} = \alpha \leq \frac{1/2 - 4\gamma\mathcal{L}}{2\gamma\mathcal{L}^2 + \frac{2}{\gamma} + L - \mu}.$$

For $\gamma \leq \frac{1}{16\mathcal{L}}$, the above inequality holds if $\omega = \mathcal{O}(\gamma\mu)$. If $\gamma = \frac{1}{16\mathcal{L}}$, the condition on variance parameter $\omega$ becomes $\omega = \mathcal{O}(\mu/\mathcal{L})$. Thus, by assumption on $\omega$ and $\gamma$, we have $2\gamma A + \alpha(L-\mu) \leq 1/2$, and hence

$$\mathbb{E}_t\left[\left\|y^{t+1} - x^*\right\|^2\right] \leq (1 - \gamma\mu)\left\|y^t - x^*\right\|^2 + D,$$

where

$$D = 2\gamma^2 B + \gamma(L-\mu)\nu + \frac{2\gamma\lambda^2}{\mu}\left\|\nabla f(\theta)\right\|^2 + 4\gamma^2\mathbb{E}\|\nabla f_\xi(x^*) - \lambda\nabla f_\xi(\theta)\|^2$$

$$= 2\gamma^2 B + \gamma(L-\mu)\nu + \frac{2\gamma}{\mu}N_3(\lambda),$$

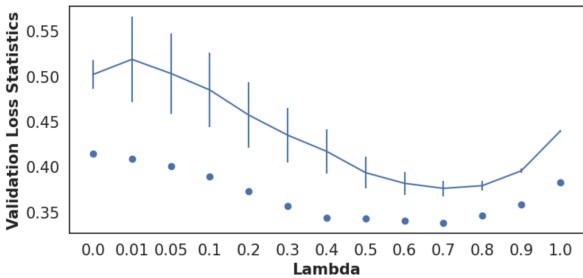

Figure 4: Validation loss statistics for the same setup as in Figure 2a.

with $N_3(\lambda) = \lambda^2 \|\nabla f(\theta)\|^2 + 2\gamma\mu\mathbb{E}\|\nabla f_\xi(x^*) - \lambda\nabla f_\xi(\theta)\|^2$. Applying Lemma 1 with $c = 2\mu$ we get $N_3(\lambda^*) = N_3(0) \min\left(1, \frac{1}{\gamma}\mathcal{O}(f(\theta) - f^*)\right) = 2\mu\sigma_*^2 \min\left(\gamma, \mathcal{O}(f(\theta) - f^*)\right)$. Taking expectation, unrolling the recurrence, and applying the tower property, we get

$$\mathbb{E}\left[\left\|y^t - x^*\right\|^2\right] \le (1 - \gamma\mu)^t \left\|y^0 - x^*\right\|^2 + \frac{D}{\gamma\mu}$$

where the neighborhood is given by

$$\begin{aligned}
\frac{D}{\gamma\mu} &= \frac{1}{\gamma\mu}(2\gamma^2 B + \gamma(L-\mu)\nu + \frac{2\gamma}{\mu}N(\lambda^*)) = \frac{1}{\gamma\mu}(2\gamma^2(\mathcal{L}^2 + 1/\gamma^2)\nu + \gamma(L-\mu)\nu + \frac{2\gamma}{\mu}N(\lambda^*)) \\
&= \frac{\nu}{\mu}(2\gamma\mathcal{L}^2 + 2/\gamma + L - \mu) + \frac{2}{\mu^2}N(\lambda^*) = \frac{\nu}{\mu}(2\gamma\mathcal{L}^2 + 2/\gamma + L - \mu) + \frac{4\sigma_*^2}{\mu}\min\left(\gamma, \mathcal{O}(f(\theta) - f^*)\right).
\end{aligned}$$

Ignoring the absolute constants and using bounds $\gamma \le \frac{1}{16\mathcal{L}}$ and $\omega = \mathcal{O}(\gamma\mu)$, we have

$$\mathbb{E}\left[\left\|y^t - x^*\right\|^2\right] \le (1 - \gamma\mu)^t \left\|x^0 - x^*\right\|^2 + \mathcal{O}\left(\frac{\omega}{\gamma\mu}\|x^*\|^2\right) + \frac{4\sigma_*^2}{\mu}\min\left(\gamma, \mathcal{O}(f(\theta) - f^*)\right).$$

Applying this inequality in (38) we conclude the proof.[2]

## F    Additional Experimental Validation

In this section we provide additional experimental validation for our theory.

### F.1    The impact of the learning hyperparameters

We begin by measuring the impact that the optimization parameters, in particular the step size, have on the convergence of SGD and KD. For this, we perform experiments on linear models trained on the MNIST dataset, without momentum and regularization, using a mini-batch size of 10, for a total of 100 epochs. We compute the cross entropy train loss between the student and true labels, measured as a running average over all iterations within an epoch, similar to Figure 2. We also compute the minimum train loss, as well as the average and standard deviation across the last 10 epochs. The results in Figure 5a show the impact that different learning rates have on the overall training dynamics of self-distillation. In all cases, the teacher was trained using the same hyperparamters as the self-distilled models ($\lambda = 0$ in the plot). We can see that using a higher learning rate introduces more variance in the SGD update, and KD would have a more pronounced variance reduction effect. In all cases, however, we can find an optimum $\lambda$ achieving a lower train loss compared to SGD.

---

[2]The rate in Theorem 4 uses $\gamma = \frac{1}{16\mathcal{L}}$ value for the learning rate.

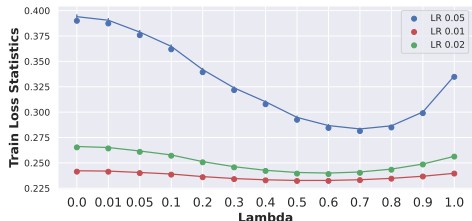 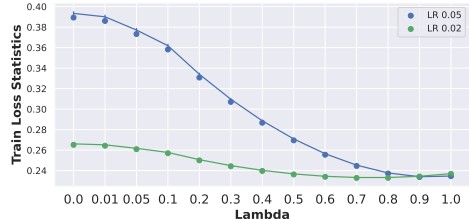

(a) The impact of the step size on the learning dynamics of SGD and KD. The teacher was trained with the same hyperparameters as the corresponding self-distilled models.

(b) The impact of a better teacher on the learning dynamics of self-distillation. The same teacher was used in both setups.

Figure 5: Ablation study on the training loss of self-distillation and SGD, when taking into account the training hyperparameters (learning rate) and quality of the teacher.

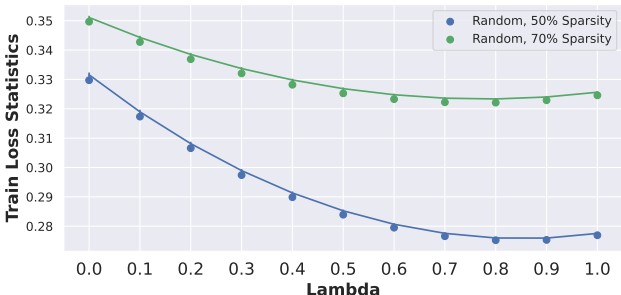

Figure 6: The impact of self-distillation on the train loss of compressed models. Here a random mask is computed at initialization, and kept fixed throughout training.

### F.2 The impact of the teacher's quality

Next, we quantify the impact that a better trained teacher could have on self-distillation. Using the same setup of convex MNIST as above, we perform self-distillation using a better teacher, i.e. one that achieves $93.7\%$ train accuracy and $92.5\%$ test accuracy. (In comparison, the teacher trained using a step size of 0.05 achieved $92\%$ train accuracy and $90.9\%$ test accuracy) We can see in Figure 5b that this better-trained teacher has a more substantial impact on the models which inherently have higher variance, i.e. those trained with a higher learning rate; in this case, the optimal value of $\lambda$ is closer to 1, which is also suggested by the theory (see Equation 14). We note that a similar behavior was also observed on the CIFAR-10 features linear classification task presented in Figure 2b.

### F.3 The impact of knowledge distillation on compression

Now we turn our attention towards validating the theoretical results developed in the context of knowledge distillation for compressed models, presented in Section 6. We consider again the convex MNIST setting, as described in the previous section, and we perform self-distillation from the better trained teacher. We prune the weights at initialization, using a random mask at a chosen sparsity level, and we apply this fixed mask after each parameter update. The results presented in Figure 6 show that self-distillation can indeed reduce the train loss, compared to SGD, even for compressed updates. Moreover, we observe that with increased sparsity the impact of self-distillation is less pronounced, as also suggested by the theory.

## G Limitations

Lastly, following our discussion from Section 7, we discuss some limitations of our work.

- As a theoretical paper, we used several assumptions to make our claims rigorous. However, one can always question each assumption and extend the theory under certain relaxations.

Our theoretical claims are based on strong (quasi) convexity or Polyak-Łojasiewicz condition, which are standard assumptions in the optimization literature.

- Another limitation concerning the "distillation for compression" part of our theory is the unbiasedness condition $\mathbb{E}\left[\mathcal{C}(x)\right] = x$ in Assumption 4. Ideally, we would utilize any "biased" compression operator, such as TopK, with similar convergence properties. However, it is known that biased estimators (e.g., biased compression operators or biased stochastic gradients) are harder to analyze in theory.

