# OpenReview forum: "Knowledge Distillation Performs Partial Variance Reduction"
_NeurIPS.cc/2023/Conference — NeurIPS 2023 poster_

### Official Review · Reviewer_ASQK · 2023-06-29

**Soundness:** 3 good
**Presentation:** 4 excellent
**Contribution:** 3 good
**Rating:** 7
**Confidence:** 3

**Summary:**

This work explores knowledge distillation, a technique used to improve the performance of smaller "student" models by leveraging the knowledge of more powerful "teacher" models. The authors analyze knowledge distillation from an optimization perspective and reveal that it can be seen as a stochastic variance reduction mechanism, reducing stochastic gradient noise and acting as a partial variance reduction technique. However, complete noise elimination depends on the characteristics of the "teacher" model. The study highlights the importance of careful parameterization, particularly regarding the weighting of the distillation loss. Empirical experiments with linear models and deep neural networks validate the findings, providing further insight into knowledge distillation.

**Strengths:**

This paper took a common empirical practice in the field, formalized the problem and provided analytical explanation from a unique perspective. The theoretical conclusions match with what the community observes in the real life, which provide valuable insights and guidance.
The experiments also perfectly corroborated the theoretical results, which makes the conclusions very convincing.

**Weaknesses:**

1. There is no explanation why LBFGS teacher is better than SGD teacher.
2. The optimal choice of distillation weight $\lambda$ does reflect the quality of the teacher, however, the selection itself is rather post hoc, is there any other more practical indicator of the quality of the teacher models that is more informative before running the optimization process.

**Questions:**

N/A

**Limitations:**

would be interesting if there are also results concerning generalizations.

---

> ### Author Rebuttal · Authors · 2023-08-05
>
> Thank you for your comments and positive evaluation of our work.
>
> - *There is no explanation why LBFGS teacher is better than SGD teacher.*
>
> ***Response.*** This is simply because LBFGS as an optimization algorithm can train the teacher to achieve almost zero train loss, while the SGD teacher converges only within some neighbourhood of the optimal solution. Throughout, for us, having a better teacher means that the loss $f(\theta) - f^*$ is smaller.
>
> - *The optimal choice of distillation weight  does reflect the quality of the teacher, however, the selection itself is rather post hoc, is there any other more practical indicator of the quality of the teacher models that is more informative before running the optimization process.*
>
> ***Response.*** This is correct. From the practical perspective, our results could be interpreted in (at least) two ways:
>
> *(1)* Our characterization shows that one could build a “line search”-type procedure to approximate the optimal choice of distillation weight for a given teacher.
>
> *(2)* More broadly, our analysis suggests that careful tuning of the distillation weight parameter can be crucial to obtain the full benefits of KD.
>
>
> - *Would be interesting if there are also results concerning generalizations.*
>
> ***Response.*** Indeed, results on generalization would be interesting, and we plan to investigate this in future work. However, in this work we take an optimization view of KD, which means that we focus on training loss.

---

> > ### Comment · Reviewer_ASQK · 2023-08-18
> >
> > Thank you for responding to my questions. Very neat paper! Keeping the previous score.

---

### Official Review · Reviewer_AWoB · 2023-07-02

**Soundness:** 4 excellent
**Presentation:** 3 good
**Contribution:** 3 good
**Rating:** 6
**Confidence:** 3

**Summary:**

This paper gives a new interpretation of the logit-based knowledge distillation algorithm. In particular, for simple one-layer models, authors show theoretically that the knowledge distillation can be viewed as diminishing the scale of the student model gradient by some amount proportional to the teacher model gradient. For deeper models, one gets a slightly different quantity, which is empirically well-correlated with the single-layer version. Building on this observation, the authors give the convergence guarantee of the knowledge distillation algorithm, in terms of the parameter and the risk (Theorem 1 and 2, respectively). Authors also propose a slight refinement of the KD, which performs a more proper form of variance reduction.

**Strengths:**

- This paper gives one of the most pleasantly simple yet insightful interpretations of the KD objective. There have been many theoretical studies that attempted to explain why knowledge distillation helps, but I do not think I have seen any explanation as simple as this. As it offers parameter-level insights, I believe that this observation can lead to many algorithmic consequences in future work.

- The theoretical analysis also provides some idea about determining the distillation weight $\lambda$, which is not practical in its current form but could give some inspiration nevertheless.

- The contributions and the proposed unbiased KD are novel, as far as I know.

- The paper is written quite clearly, and the contributions are easy to understand.

**Weaknesses:**

- It would have been better if the paper provides a direct head-to-head comparison of theorems 1 & 2 to the convergence guarantees one could get from the vanilla SGD procedure. The provided "convergence bound" type of results does not really give too much insight into how the knowledge distillation (or variance reduction) leads to a **better** convergence than the vanilla SGD.

- Two logical connections could be made a little bit clearer. (1) How does eq(5) lead to standard variance reduction? Are we saying that KD is a variance reduction, only because there are some negative terms? Do we have any direct empirical corroborations of this claim? (2) Please explain how variance reduction leads to better training, e.g., by citing prior work.

- In Figure 1, I am not sure whether the cosine similarity is the right metric to use. The magnitude should be a very critical issue, especially because this paper is claiming that KD is about variance reduction. Could you provide additional plots on the $\ell_2$ difference, or maybe the SNR-like result (i.e., the ratio between the difference and the original distillation gradient)?

- The empirical results on the proposed unbiased KD are given in the form of "training loss" only, rather than test loss or test accuracy. The results on the test/validation dataset may be very useful in understanding the strengths and limitations of the proposed algorithm.

- A mild suggestion on notations---please consider changing the symbol for the student model parameters. In most machine learning papers, $x \in \mathcal{X}$ denotes the input feature (and $y \in \mathcal{Y}$ denotes the label). Many KD papers use $\theta_t$ for teacher parameters and $\theta_s$ for student parameters. Although the current notation is okay, using more common notations may help the readers greatly.

- A minor mistake---the example on "classification with single hidden layer" may not really be the case with one "hidden layer." The one-hidden-layer network is the same as a two-layer neural network.

**Questions:**

In addition to the "weaknesses" above, here are some questions/suggestions.

- Is there any way one could extend the proposed method of analysis to the distillation cases where the student model is not necessarily a "compressed" version of the teacher model?

- It would be great if there is a plot that explicitly tracks the gradient variance.

**Limitations:**

The limitations are given in the appendices, which I think is okay.

---

> ### Author Rebuttal · Authors · 2023-08-05
>
> Thank you for your comments and positive evaluation of our work
>
> - *It would have been better if ...*
>
> ***Response.*** We acknowledge this suggestion, and in fact this is one of the key aspects of our theory which is discussed in lines 269-277 (Importance of the results) after the proof overview of the theorems. The comparison of the convergence rates between theorems 1 & 2 and SGD can be done as follows. For both setups, the rate of SGD is the same (11) or (12) with only one difference: $\min(\gamma, \mathcal{O}(f(\theta) - f^*))$ term is replaced with $\gamma$. So, $\mathcal{O}(f(\theta) - f^*)$ is the factor that makes our results ***better*** compared to SGD in terms of optimization performance.
>
> We thank you for this suggestion, and will add further clarifying discussion on this point.
>
> - *Two logical connections could be made a little bit clearer...*
>
> ***Response.*** Thank you for the suggestion, we will make them clearer in the revision.
>
> *(1)* Exactly! Equation (5) leads to partial VR because of the additional $ -\lambda\nabla f_{\xi}(\theta)$ term. When chosen properly (distillation weight $\lambda$ and proximity of $\theta$ to the solution $x^*$), this additional stochastic gradient is capable of adjusting the student’s stochastic gradient since both are computed using the same batch from the train data. In other words, both gradients have the same source of randomness which makes partial cancellations feasible.
> As a matter of fact, we do have direct empirical corroboration for this in Figure 3 (see also the new plot, Figure 7 in our response PDF, tracking gradient variance for the same setup). The Blue line is pure SGD, without any distillation involved. The Green line employs distillation with distillation weight $\lambda=0.4$ and achieves uniformly lower train errors (and gradient variance).
>
> *(2)* First of all, in the optimization perspective we adopt, “better training” means precisely smaller train error. The analysis of test error is a different aspect which we do not consider here. The standard/complete variance reduction mechanism (such as SVRG) is capable of removing the stochastic neighborhood term completely, and guarantees that the convergence is to the exact minimizer, as opposed to SGD. To give a specific example, in the strongly convex and smooth regime SGD with constant learning rate still converges up to some neighborhood of the minimizer, while SVRG converges to the exact solution with constant learning rate. See reference [4] below for a complete discussion.
>
> [4] Robert M. Gower, Mark Schmidt, Francis Bach, Peter Richtarik, *Variance-Reduced Methods for Machine Learning*, IEEE, 2020 (https://arxiv.org/abs/2010.00892).
>
> We will provide a clarifying discussion on both points in the next version of our paper.
>
> - *In Figure 1, I am not sure whether the cosine similarity is the right metric to use. ...*
>
> ***Response.*** We believe that cosine similarity is a relevant metric since it measures the alignment of the gradients. Nevertheless, to fully address this concern, we additionally provided plots tracking $\ell_2$ distance and SNR for the same gradients in the rebuttal PDF (Figure 6). As we can see, they follow exactly the same trend as the plot of cosine similarity.
>
>
> - *The empirical results on the proposed unbiased KD are given in the form of "training loss" only, rather than test loss ...*
>
> ***Response.*** Please note that the purpose of proposing / analyzing unbiased KD is to support our claim that the potential source of ***partial*** variance reduction in KD is the biased nature of distillation. Indeed, we removed the biased and improved variance reduction both analytically and empirically. Analyzing test loss/accuracy of unbiased KD would be interesting in itself, but is irrelevant for our argument related to variance reduction. In fact, unbiased KD closely relates to the popular and well-studied SVRG algorithm (please see the references in the paper), applied in the context of deep networks.
>
> - *A mild suggestion on notations*
> - *A minor mistake*
>
> ***Response.*** Thank you for the detailed comments on improving the presentation, which we will take into account for the next version of the work.
>
> - *Is there any way one could extend the proposed method of analysis to the distillation cases where the student model is not necessarily a "compressed" version of the teacher model?*
>
> ***Response.*** Yes, we believe it is possible to extend the current method of analysis to general knowledge distillation under certain restrictions. The main issue there is obtaining reasonably accurate analytical expressions for the distillation gradients. If the student’s architecture is not the same as the teacher’s nor is a sub-network of the teacher’s architecture, then it may be tricky to derive a closed form for the distillation gradient. Technically, the gradient space of the teacher could be very different from the gradient space of the student; dimensions may not match and the student's gradient space is not a subspace of the teacher's gradient space. One possible workaround could be to impose additional assumptions on how the student’s gradient space can be embedded reasonably into the teacher’s gradient space. (For instance, this would be the case with pruning, quantization, or structured compression/neural architecture search.) In such cases, we will indeed be able to extend Proposition 1.
>
> - *It would be great if there is a plot that explicitly tracks the gradient variance.*
>
> ***Response.*** Please find the plot in our PDF response (Figure 7). The plot explicitly tracks gradient variance (averaged over the iterations within each epoch) for the same setup as Figure 3. As expected, both variants of KD (biased and unbiased) have reduced gradient variance compared to plain SGD. The plot also highlights that both variants of KD have similar variance reduction properties, while the unbiasedness of unbiased KD amplifies the reduction of train loss in Figure 3.

---

> > ### Comment · Reviewer_AWoB · 2023-08-16
> >
> > I very much appreciate the careful comments and additional results. My concerns have been addressed well.

---

### Official Review · Reviewer_6fjL · 2023-07-02

**Soundness:** 2 fair
**Presentation:** 3 good
**Contribution:** 3 good
**Rating:** 6
**Confidence:** 3

**Summary:**

This paper examines the benefits of Knowledge Distillation (KD) from an optimization perspective. They show that, under certain assumptions, KD performs partial variance reduction on SGD noise and that the amount of reduction depends on the quality of the teacher model. Their analysis suggests that the distillation weight used in the KD loss should be appropriately tuned, and the authors provide a closed-form solution for the optimal weight in the case of linear models. Even though their core result does not directly apply to deep networks, they present some empirical evidence supporting that it remains a reasonable approximation.

**Strengths:**

* Understanding the underlying mechanics of KD and the reason for its benefit is highly relevant to the ML community given the widespread use of KD. A deeper understanding may also lead to better distillation algorithms, as exemplified with the closed-form optimal distillation weight in this paper.
* The connection between the distillation gradient and variance reduction methods like SVRG is interesting and novel to the best of my knowledge.
* The presented analyses and results seem sound.
* The authors present some empirical evidence that support the claims of the paper (Figs. 2-5).


**Weaknesses:**

* The core proposition of the paper (Prop. 1) does not apply to deep neural networks, and the empirical evidence presented to support the claim that it’s a good approximation is quite limited (one scenario on MNIST with one hidden layer FFN and fixed learning rate).
* The presented results, including the empirical ones, apply to training loss, rather than test error.
* The presented theory states that a higher performing teacher leads to higher variance reduction. However, prior works in distillation have shown that significantly better teachers often lead to worse students [1-3]. Please see the questions section for specific questions on this.

[1] https://arxiv.org/pdf/1902.03393.pdf

[2] https://arxiv.org/pdf/2202.03680.pdf

[3] https://arxiv.org/pdf/2206.06067.pdf

**Questions:**

1. Continuing from the comment from the Weakness section: how does the presented theory reconcile with the prior observations that better teachers may in fact lead to poorer students due to, e.g., the large capacity gap between the student and the teacher? Does this suggest that variance reduction is not the end of the story or that it is solely responsible for the success of distillation?

2. Why does the cosine similarity between the approximation of Prop 1 and the true gradient in deep networks decrease as the training progresses? It would be helpful to show the behavior until a larger epoch like 100 instead of 50 to support the claim that the behavior stabilizes as training progresses.

3. Beyond empirical evidence, is there an intuitive reason to believe that Prop. 1 would approximately apply to deep neural networks?


**Limitations:**

Yes.

---

> ### Author Rebuttal · Authors · 2023-08-05
>
> Thank you for your comments and evaluation of our work.
>
>
> - *The core proposition of the paper (Prop. 1) does not apply to deep neural networks, and the empirical evidence presented to support the claim that it’s a good approximation is quite limited (one scenario on MNIST with one hidden layer FFN and fixed learning rate).*
>
> ***Response.*** We acknowledge the fact that the focus of our work is analytical: we identify a first non-trivial interpretation of KD from the optimization perspective, and characterize cases when KD can lead to better rates. Our interpretation appears to match experiments precisely in the convex case. Our further experiments are meant to show that our “distillation gradient” interpretation can also be relevant in the case of SGD-based optimization of DNNs. We indeed plan to investigate the challenging DNN / non-convex case more precisely in future work, for which both new techniques and further experimental validation will be needed.
>
> - *The presented results, including the empirical ones, apply to training loss, rather than test error.*
>
> ***Response.*** As we also mentioned in the abstract, in this work, we investigate KD specifically from an optimization perspective. Of course, investigating test errors is also important. However, it is not the goal of this work. Nonetheless, we added a plot (Figure 8 in the response PDF file) showing validation loss for the same setup as Figure 2(a) with very similar behavior.
>
>
> - *1. Continuing from the comment from the Weakness section: how does the presented theory reconcile with the prior observations that better teachers may in fact lead to poorer students due to, e.g., the large capacity gap between the student and the teacher? Does this suggest that variance reduction is not the end of the story or that it is solely responsible for the success of distillation?*
>
> ***Response.*** To reconcile the mentioned prior observations with our results, notice that the notion of a “better teacher” has different meanings. In the papers you mentioned, a better teacher means a teacher with a much larger capacity (for instance wider and/or deeper architecture) which can have higher performance than the student model. However, in our case “better teacher” means better parameter (i.e., weights and biases) values, evaluated in terms of training loss.
>
> In particular, in the case of self-distillation, covered in Sections 4 and 5, the teacher and student architectures are identical, and hence they have the same capacity. Here, a better teacher means better parameter values within the same architecture.
>
> In our second regime, distillation for compressed models (Section 6), we actually consider the case when the student network is a subnetwork of the teacher; we consider a sparsification compression operator that selects $k$ parameters for the student out of $d$ parameters of the teacher. Then, clearly, the teacher has a larger capacity with a capacity ratio $d/k\ge1$. However, our result in this direction (Theorem 4) does not allow the capacity ratio to be arbitrarily large. Indeed, the constraint $\omega = \mathcal{O}(\mu/\mathcal{L})$ on compression variance implies a constraint on capacity ratio since $\omega = d/k-1$ for the sparsification operator. Thus, our result holds when the teacher’s size is not significantly larger than the student’s size, which does not contradict the observations from prior work noted by the reviewer.
>
>
> - *2. Why does the cosine similarity between the approximation of Prop 1 and the true gradient in deep networks decrease as the training progresses? It would be helpful to show the behavior until a larger epoch like 100 instead of 50 to support the claim that the behavior stabilizes as training progresses.*
>
> ***Response.*** Please see our PDF response where we added the plot (Figure 6) with a larger 100 epoch training supporting the claim on stability. The decrease of cosine similarity can be explained as follows: at the beginning the cosine similarity is high (and SNR is low) since we start from the same model. Then, initial perturbations caused by either the KD or modified KD gradient don’t cause big shifts (the teacher has enough confidence and small gradients). These perturbations accumulate over the training leading to decreased cosine similarity and eventually stabilize.
>
> - *3. Beyond empirical evidence, is there an intuitive reason to believe that Prop. 1 would approximately apply to deep neural networks?*
>
> ***Response.*** Yes, there is. Intuitively, distillation has essentially negligible effect on the data points for which the teacher classifies correctly with high confidence (since the distillation gradient almost equals the regular gradient). In such cases, the feedback from the teacher is similar to or the same as from the true labels. This intuition is reflected in Proposition 1 since high confidence of teacher’s classification means that teacher’s gradient $\nabla f_n(\theta)$ with respect to those data points is close to zero and does not affect the student’s gradient much.

---

> > ### Comment · Reviewer_6fjL · 2023-08-10
> >
> > Thank you for your response. I read the authors' rebuttal, the general response, and the other reviews. In light of the new results in the PDF provided in the PDF and the author's promise to provide a more complete discussion towards applicability in the non-convex case in the next revision, I decided to raise my score. I still view the fact that the theory predicts that a more capable teacher (in terms of lower train error, which is generally the case with a higher capacity teacher) is contradictory to some of the empirical observations from prior work. This could be due to the fact that the bounds are strictly from an optimization perspective, which concerns training performance. I would encourage the authors to include a discussion on this to make this explicit in light of prior work showing higher capacity teachers may not necessarily lead to better students (which seems to be implied by the presented theory at a first glance).

---

> > > ### Author Response · Authors · 2023-08-11
> > >
> > > Thank you for your response and the score increase. We acknowledge your point, and we will provide a detailed discussion on the possible interpretations of our results in light of the existing empirical work on KD, both in terms of training and generalization performance.
> > >
> > > Generally, we believe there is no contradiction between our results and the existing empirical work on KD. Our focus in this work is solely on training performance (e.g., better student/teacher means better parameter values for train error), while the empirical observations (following the three papers you mentioned in your initial review, which we examined) are in terms of generalization performance. Moreover, none of our results consider the regime where the teacher’s capacity (in terms of architecture size) is significantly larger than that of the student.

---

### Official Review · Reviewer_1HYD · 2023-07-13

**Soundness:** 3 good
**Presentation:** 3 good
**Contribution:** 3 good
**Rating:** 7
**Confidence:** 3

**Summary:**

This work analyzes Knowledge Distillation (KD) using an optimization point of view.
By recasting the KD problem as a standard learning problem with a custom loss, the authors analyze the convergence of SGD on such loss and identify the variance-reducing properties of KD, through the bias induced by the teacher in the loss.
Using some approximations, the authors argue that their analysis should hold for deep networks too.

The paper complements its analysis by proposing a technique to reduce the bias of KD and extend the result to generic KD with smaller students.
The proposes analysis is supported by experiments for linear models.

**Strengths:**

The paper is clear and easy to read even for non-optimization experts.
The whole idea of analyzing KD as a variance-reducing algorithm is novel and interesting
Although the assumptions are limited to linear models, the empirical experiments show that such assumptions approximately could hold deep networks is convincing.
The paper proposes a good balance of novelty, clarity and significance and it should be relevant to anybody interested in KD.

**Weaknesses:**

The biggest weakness of this work is its ambivalence on the validity of the results for deep learning.
While I do understand this is not the goal of the paper, I do not understand why the authors claim that their results could be approximately true for deep networks, show some empirical evidence of this and then, drop the topic for the rest of the paper.
I don't think the authors could either add some empirical evidence on this matter or focus on the linear/convex case.
I any case, I encourage the authors to further investigate KD as variance-reductions on deep networks, even if only empirically in another paper.

**Questions:**

My doubts are about the most empirical aspects of the paper. The "Experimental validation" in sec 4.3 mentions "linear probing" on CIFAR-10, is this just taking the last features produced by a neural network and training a linear classifier on top of those?

I also want to ask to the authors if they tried to replicate experiments with deep networks to show that empirically, the same behaviours can be obtained even for non-linear models.
I don't understand why you write a paragraph to claim that the fundamental assumption for your results can hold approximately for deep learning and then don't elaborate on the matter any further, either with experiments or theoretical analysis.

**Limitations:**

The authors have addressed the limitations of their work properly.

---

> ### Author Rebuttal · Authors · 2023-08-05
>
> Thank you for your comments and positive evaluation of our work.
>
>
> - *The biggest weakness of this work is its ambivalence on the validity of the results for deep learning. ...*
>
> ***Response.*** We acknowledge the fact that the focus of our work is analytical: we identify a first non-trivial interpretation of KD from the optimization perspective, and characterize cases when KD can lead to better rates. Our interpretation appears to match experiments precisely in the convex case. Our further experiments are meant to show that our “distillation gradient” interpretation can also be relevant in the case of SGD-based optimization of DNNs. We indeed plan to investigate the challenging DNN / non-convex case more precisely in future work, for which both new techniques and further experimental validation will be needed.
>
> - *My doubts are about the most empirical aspects of the paper. The "Experimental validation" in sec 4.3 mentions "linear probing" on CIFAR-10, is this just taking the last features produced by a neural network and training a linear classifier on top of those?*
>
> ***Response.*** Yes, we train a linear classifier on top of the features extracted from a ResNet50 model pre-trained on ImageNet. This is a standard setting, commonly used in the transfer learning literature, see e.g. [1], [2], [3].
>
> [1] Simon Kornblith, Jonathon Shlens, Quoc V. Le, *Do Better ImageNet Models Transfer Better?* CVPR, 2019 (https://arxiv.org/abs/1805.08974)
>
> [2] Hadi Salman, Andrew Ilyas, Logan Engstrom, Ashish Kapoor, Aleksander Madry, *Do Adversarially Robust ImageNet Models Transfer Better?* NeurIPS, 2020 (https://arxiv.org/abs/2007.08489)
>
> [3] Eugenia Iofinova, Alexandra Peste, Mark Kurtz, Dan Alistarh, *How Well Do Sparse Imagenet Models Transfer?* CVPR, 2022 (https://arxiv.org/abs/2111.13445)
>
>
> - *I also want to ask to the authors if they tried to replicate experiments with deep networks to show that empirically, the same behaviours can be obtained even for non-linear models. I don't understand why you write a paragraph to claim that the fundamental assumption for your results can hold approximately for deep learning and then don't elaborate on the matter any further, either with experiments or theoretical analysis.*
>
> ***Response.*** We acknowledge that only provide partial evidence for the validity of our “distillation gradient” assumption for deep models. We provided evidence that the assumption holds approximately for deep networks, and show varying behaviour of the alignment during training.
> Further, we emphasize that our Theorem 2 is designed to hold specifically for self-distillation in the non-convex case, which approximates very popular heuristics in practice, e.g. [55].
>
> We agree that there is much more to investigate for the case of deep networks where exact tracking of teacher’s impact across multiple layers of non-linearities becomes harder. We see our results as a promising first step towards a more complete understanding of the effectiveness of distillation, and will provide a more complete discussion towards applicability in the non-convex case in the next revision.

---

### Author Rebuttal · Authors · 2023-08-05

Dear Reviewers and Area Chair,

Thank you for the time and effort you put into evaluating our work. We have responded to all comments in your reviews by providing additional discussions/clarifications and numerical validations (please find the attached PDF response for the additional plots).

Please let us know whether we managed to address your concerns regarding the paper.

Regards,
Authors

---

### Decision · Program_Chairs · 2023-09-21

**Decision:**

Accept (poster)

**Comment:**

This works studies knowledge distillation through the optimization lens and shows that distillation can lead to variance reduction of SGD. The paper presents a theoretical analysis of deep linear models, and empirically verifies that the predictions of the theory carry over to general deep networks. All reviewers find the connection between distillation and variance reduction interesting and novel and they commonly recommend accept. A shared shortcoming of this work mentioned by the reviewers is that the linear network analysis is not enough to guarantee similar results always hold true for general neural networks . However, they also agree that the paper even has enough contribution to be published. In concordance with the reviewers, I recommend accept. Please incorporate the clarifications that you provided during the rebuttal into the final version of the paper.